# Intercomparison of Middle Atmospheric Meteorological Analyses for the Northern Hemisphere Winter 2009-2010

John P. McCormack[1,†], V. Lynn Harvey[2,3], Cora E. Randall[2,3], Nicholas Pedatella[4], Dai Koshin[5,8], Kaoru Sato[5], Lawrence Coy[6,7], Shingo Watanabe[8], Fabrizio Sassi[1], Laura A. Holt[9]

[1]Space Science Division, Naval Research Laboratory, Washington DC, USA
[2]Laboratory for Atmospheric and Space Physics, University of Colorado, Boulder CO, USA
[3]Department of Atmospheric and Oceanic Sciences, University of Colorado, Boulder CO, USA
[4]High Altitude Observatory, National Center for Atmospheric Research, Boulder CO, USA.
[5]Department of Earth and Planetary Science, The University of Tokyo, Tokyo, Japan
[6]Science Systems and Applications, Landover MD, USA
[7] NASA Goddard Space Flight Center, Greenbelt MD, USA
[8]Japan Agency for Marine-Earth Science and Technology, Yokohama, Japan
[9]Northwest Research Associates, Boulder CO, USA
[†]Now at Heliophysics Division, Science Mission Directorate, NASA Headquarters, Washington DC, USA

*Correspondence to*: V. Lynn Harvey (lynn.harvey@lasp.colorado.edu)

**Abstract.** Detailed meteorological analyses based on observations extending through the middle atmosphere (~15 to 100 km altitude) can provide key information to whole atmosphere modelling systems regarding the physical mechanisms linking day-to-day changes in ionospheric electron density to meteorological variability near the Earth's surface. However, the extent to which independent middle atmosphere analyses differ in their representation of wave-induced coupling to the ionosphere is unclear. To begin to address this issue, we present the first intercomparison among four such analyses, JAGUAR-DAS, MERRA-2, NAVGEM-HA, and WACCMX+DART, focusing on the Northern Hemisphere (NH) 2009-2010 winter, which includes a major sudden stratospheric warming (SSW). This intercomparison examines the altitude, latitude, and time dependences of zonal mean zonal winds and temperatures among these four analyses over the 1 December 2009 – 31 March 2010 period, as well as latitude and altitude dependences of monthly mean amplitudes of the diurnal and semidiurnal migrating solar tides, the eastward propagating diurnal zonal wave number 3 nonmigrating tide, and traveling planetary waves associated with the quasi-5 day and quasi-2-day Rossby modes. Our results show generally good agreement among the four analyses up to the stratopause (~50 km altitude). Large discrepancies begin to emerge in the mesosphere and lower thermosphere owing to (1) differences in the types of satellite data assimilated by each system and (2) differences in the details of the global atmospheric models used by each analysis system. The results of this intercomparison provide initial estimates of uncertainty in analyses commonly used to constrain middle atmospheric meteorological variability in whole atmosphere model simulations.

## 1 Introduction

The atmospheric region from approximately 15 to 100 km altitude spanning the stratosphere, mesosphere, and lower thermosphere is often referred to as the "middle atmosphere". Through recent advances in numerical modelling and data assimilation capabilities, it is now understood that the middle atmosphere plays an important role in determining

how meteorological variability near the Earth's surface affects the state of the coupled thermosphere/ionosphere (T/I) system (~100 to 500 km altitude) on time scales from hours to months. In addition to the well-established solar and geomagnetic drivers of the T/I system, this meteorological variability can impact the performance of space-based geolocation and global communication systems, and this impact is particularly noticeable during times of reduced solar activity. Specifically, these space-based systems are affected by rapid changes in the ionospheric electron content, which is determined by a complex interplay between variations in the thermospheric density, chemical composition, and circulation, particularly in the dynamo region of the thermosphere from 100 to 200 km that includes the ionospheric E and lower F regions.

Figure 1 illustrates examples of internal drivers of T/I variability, including planetary scale waves, gravity (or buoyancy) waves, and tides that are produced in the troposphere and stratosphere and propagate upward through the middle atmosphere. The present study focuses on how some basic characteristics of these drivers are represented in meteorological analyses that extend throughout the middle atmosphere, as this critical altitude region can be viewed as the conduit between meteorological variability near the surface and related changes in the T/I system. Coupling between the state of the middle atmosphere and the behavior of the T/I system has been demonstrated in observational studies (e.g., Goncharenko and Zhang, 2008; Chau et al., 2009; Goncharenko et al., 2010; Pedatella and Forbes, 2010) linking variations in total electron content and ion drift with the reversal of polar stratospheric flow in the Northern Hemisphere (NH) winter during sudden stratospheric warmings (SSWs). Subsequent modelling studies showed that changes in the amplitude and phase of both migrating and nonmigrating tides are the primary drivers of changes in the T/I state in response to SSWs that result in anomalous ionospheric behaviour. However, as shown by, e.g., Pedatella et al. (2014a), whole atmosphere models produce widely varying estimates of the tidal variability within the T/I region. The reason for this disagreement can be attributed to both differences in model physics and differences in the data sets used to constrain these models.

Differences in model physics, especially the treatment of gravity wave processes, no doubt play a role in explaining some of the inter-model discrepancies reported by Pedatella et al. (2014a) with respect to both the background zonal mean state and tidal variability within the thermosphere. The primary gravity waves illustrated in Fig. 1 are excited near the surface and propagate up, growing in amplitude and becoming unstable or "breaking" in the mesosphere, depositing heat and momentum into the background flow. Primary gravity wave breaking often occurs at spatial scales too small to be resolved in global models, and typically is represented in these models by single column parameterizations with tropospheric sources. Also shown in Fig. 1 are secondary gravity waves triggered by flow instabilities related to primary gravity wave breaking in the mesosphere, which may propagate into the lower thermosphere and drive T/I variability (Becker and Vadas, 2018; Vadas and Becker, 2018). Currently, global atmospheric models extending into the thermosphere do not account for the effects of secondary gravity wave breaking. More advanced treatments of gravity wave breaking in the mesosphere and lower thermosphere (MLT) region are thus needed to better understand and ultimately predict internal drivers of T/I variability.

Pedatella et al. (2014a) also noted that some of the models employed different meteorological analyses to constrain (or "nudge") meteorological variability in the middle atmosphere. These analyses are produced through assimilation of atmospheric observations mainly in the troposphere and stratosphere and were initially developed for a wide range of applications that include initialization and validation of numerical weather prediction (NWP) systems and long-term climate studies. Some well-known examples of these analyses include the second-generation Modern-Era Retrospective analysis for Research and Applications (MERRA-2; Bosilovich et al., 2015), the European Centre for Medium-range Weather Forecasting Interim Atmospheric Reanalysis (ERA-I; Dee et al., 2011), the National Centers for Environmental Prediction/National Center for Atmospheric Research (NCEP/NCAR) reanalysis (Kalnay et al., 1996; Kistler et al., 2001), and the Japanese Meteorological Agency's 55-year reanalysis (JRA55; Kobayashi et al., 2015). Understanding how whole atmospheric model simulations of T/I variability are impacted by the use of different meteorological reanalyses as constraints (e.g., Sassi et al., 2021) could help understand the origins of inter-model discrepancies such as those noted by Pedatella et al (2014a).

A recent intercomparison of several reanalyses was performed as part of the Stratospheric Reanalysis Intercomparison Project (S-RIP; Fujiwara et al., 2017), with a chapter focusing specifically on the ability of reanalyses to capture key processes in the upper stratosphere and lower mesosphere (Harvey et al., 2021). A key finding of Harvey et al. (2021) is that the most commonly used reanalyses (e.g., MERRA-2, ERA-I, JRA55) show good agreement in their representation of the zonal mean atmospheric state and in their representation of planetary waves (PWs) and tides up to ~50 km altitude, but the representations diverge quite substantially above 50 km altitude, particularly in the equatorial region. This is not surprising, since these systems were originally developed with a focus on tropospheric and stratospheric applications, with top levels extending into the lower mesosphere (~60 km altitude) in most cases. In addition, the lack of wind measurements at low latitudes above 10 hPa (~30 km) combined with the breakdown of midlatitude geostrophic balance adds to the analysis uncertainty in this important tidal region. However, this disagreement among reanalyses above the stratopause poses a challenge for emerging whole atmosphere modelling applications, such as those described above, that seek to quantify the response of the T/I system to meteorological variability in the middle atmosphere. For example, Sassi et al. (2018) demonstrated that whole atmosphere model simulations constrained with high-altitude meteorological analyses extending up to ~90 km altitude represented day-to-day variability in the lower thermosphere more realistically than simulations constrained with analyses that only extended up to ~60 km altitude, especially around the time of a major SSW. Constraining whole atmosphere models by using meteorological analyses with widely varying representations of the middle atmosphere state above ~60 km altitude makes it difficult to conclusively identify and predict the physical drivers that are responsible for linking lower atmospheric meteorology to ionospheric variability.

To address the emerging need for accurate global atmospheric analyses throughout the entire middle atmosphere, high-altitude data assimilation and modelling systems (e.g., Pedatella et al., 2014b; McCormack et al., 2017; Koshin et al., 2020) have been developed recently to provide observations-based constraints of middle atmospheric meteorological variability for whole-atmosphere models (Sassi et al., 2018; McDonald et al., 2018; Pedatella et al., 2019). These

systems produce global meteorological analyses by incorporating both standard operational meteorological observations near the surface and satellite-based observations of the middle atmosphere from dedicated NASA research missions such as Aura (Schoeberl et al., 2006) and TIMED (Thermosphere-Ionosphere-Mesosphere Energetics and Dynamics; Yee et al., 1999). Typical top levels for these new systems extend to 90 km altitude or higher, so each of these systems provides valuable resources for studying the dynamics and variability of the middle atmosphere. Examining the level of agreement among these new high-altitude systems is a first step towards understanding how whole atmosphere model simulations may be affected when constrained by different sets of meteorological input.

This paper presents the first intercomparison of four analyses extending into the middle atmosphere: the high-altitude version of the Navy Global Environmental Model (NAVGEM-HA; Eckermann et al., 2018), the Whole Atmosphere Community Climate Model with thermosphere-ionosphere eXtension using the Data Assimilation Research Testbed (WACCMX+DART; Pedatella et al., 2018), the Japanese Atmospheric General circulation model for Upper Atmosphere Research with Data Assimilation System (JAGUAR-DAS; Koshin et al., 2020; 2021); and MERRA-2. Each of these systems assimilates middle atmosphere data to varying degrees, with top output levels ranging from 80 km to ~500 km altitude. The objective of this study is to quantify the similarities and differences between these four analyses. The results are useful for assessment of uncertainty in constrained or "nudged" whole atmosphere simulations arising from differences in meteorological inputs. These results can also be used to highlight where further improvements in middle atmospheric data assimilation and modelling are needed in order to improve our understanding of how meteorological variability impacts day-to-day variability in ionospheric conditions, especially during quiet Sun conditions.

The initial plan for this intercomparison was conceived as a follow-on study of Harvey et al. (2021) by the SPARC (Stratosphere-troposphere Processes and their Role in Climate) Data Assimilation Working Group (https://www.sparc-climate.org/activities/data-assimilation/) to examine high altitude meteorological analyses extending throughout the middle atmosphere. Due to the large computational resources needed to generate these types of meteorological analyses, a detailed multi-year intercomparison is not currently within the scope of the present study. Instead, we focus on a detailed examination of the four analyses over the 1 December 2009 – 31 March 2010 period, which includes a major SSW. This work is particularly interested in mesospheric wind and temperature disturbances that occur in late January (Goncharenko et al., 2013; Jones et al., 2018; McCormack et al., 2017), two weeks before the onset of easterlies in the stratosphere on 9 February (Butler et al., 2017). This Northern Hemisphere (NH) wintertime period is useful since it provides a prime example of a dramatic shift in middle atmospheric circulation that has been studied extensively through both observations and modelling studies.

The paper is organized as follows: The four meteorological analyses used in this intercomparison (NAVGEM-HA, WACCMX+DART, JAGUAR-DAS, and MERRA-2) are described in section 2. Section 3 describes the numerical methods used to analyze space-time variations in the data related to specific PW and tidal features. Section 4 presents an intercomparison of the zonal mean zonal wind and zonal mean temperature data, while section 5 presents an

intercomparison of the PW and tidal signatures. The results of this study are summarized, and implications for future research are discussed, in Section 6.

## 2. Data and Methods

This section provides an overview of each of the four high-altitude meteorological systems used in the present intercomparison of the NH winter period extending from 1 December 2009 to 31 March 2010. Each of these systems combines a data assimilation (DA) component with an atmospheric model component that together produce global synoptic analyses of key atmospheric quantities. In the discussion below, we describe the main features of the DA and modelling systems relevant for capturing specific PW and tidal components; previous observational and modelling studies (see Section 1) have shown these PWs and tides can impact day-to-day variability in the T/I system. These include the migrating diurnal and semidiurnal solar tides (referred to here as DW1 and SW2, respectively), the non-migrating diurnal eastward zonal wavenumber 3 tidal component (DE3), the quasi-2-day wave (Q2DW), and the quasi-5-day wave (Q5DW).

For this intercomparison, we examine global gridded data sets of temperature, zonal wind and geopotential height from four different systems extending throughout the middle atmosphere and in some cases (JAGUAR-DAS and WACCMX+DART) into the thermosphere. The main sources of middle atmosphere observations for these systems are retrieved vertical temperature profiles from the Aura Microwave Limb Sounder (MLS; Schwartz et al., 2008) between ~16 km and 90 km altitude and extending from 82°S to 82°N latitude, and from the TIMED Sounding of the Atmosphere using Broadband Emission of Radiation (SABER; Remsberg et al., 2008) instrument between ~16 km and 105 km altitude with latitude coverage that alternated between its south viewing mode (83°S-52°N) and north-viewing mode (83°N-52°S) on 11 January 2010. Further details on each high-altitude analysis system can be found in the discussion below and references therein. All data used in this study is publicly available as described in the Acknowledgments section.

Table 1 gives overall references for each system, lists the horizontal, vertical, and temporal resolutions, gives the vertical range for the systems, and provides references for the orographic and non-orographic gravity wave parameterizations implemented in each system.

| Analysis System | Reference(s) | Horizontal Grid, Vertical Grid, and Output Frequency | Vertical Range | Reference(s) for Gravity Wave Drag Parameterizations |
|---|---|---|---|---|
| JAGUAR-DAS | Koshin et al. (2020); Koshin et al. (in review) | 2.8125° lat/lon, $\Delta z \approx 1$ km, $\Delta t = 6$ hours | Surface to $1 \times 10^{-6}$ hPa (~150 km) | ORO: McFarlane (1987) NON: Hines (1997); Watanabe (2008) |
| MERRA-2 | Bosilovich et al. (2015); Gelaro et al. (2017); Molod et al. (2015) | 0.625° lon by 0.5° lat, $\Delta z \approx 2\text{-}5$ km, $\Delta t = 3$ hours | Surface to 0.01 hPa (~75 km) | ORO: McFarlane (1987) NON: Garcia & Boville (1994); Molod et al. (2015) |
| NAVGEM-HA | McCormack et al. (2017); Eckermann et al. (2018) | 1° lat/lon, $\Delta z \approx 2\text{-}4$ km, $\Delta t = 3$ hours | Surface to $6 \times 10^{-5}$ hPa (~120 km) | ORO: Webster et al. (2003) NON: Eckermann (2011) |
| WACCMX+DART | Liu et al. (2018); Pedatella et al. (2018) | 1.9° lat by 2.5° lon, $\Delta z \approx 1\text{-}5$ km, $\Delta t = 1$ hour | Surface to $4.1 \times 10^{-10}$ hPa (~500-700 km) | ORO: McFarlane (1987) NON: Beres et al. (2005); Richter et al. (2010); Garcia et al. (2017) |

**Table 1.** List of analysis datasets used in this paper, overall references describing each system, the horizontal, vertical, and temporal characteristics of the analysis output, the model top, and references for gravity wave specifications. In the 5th column, ORO refers to the parametrization for orographic gravity waves while NON refers to that of non-orographic gravity waves.


## 2.1 NAVGEM-HA

NAVGEM-HA is a research version of the U.S. Navy's operational NWP system developed for middle atmosphere applications. It processes over 6 million atmospheric observations within its standard 6-hour assimilation window,
consisting of surface station reports, radiosondes, and numerous operational meteorological satellites (McCormack et al., 2017; Eckermann et al., 2018). In addition to MLS and SABER temperature retrievals, NAVGEM-HA also assimilates vertical profiles of ozone and water vapor from MLS, and microwave radiances from the upper atmospheric sounder (UAS) channels of the Special Sensor Microwave Imager/Sounder (SSMI/S), as illustrated in Fig. 3a of Eckermann et al. (2018). Over the 2009-2010 period of this intercomparison, three different space-based platforms
(designated F16, F17, and F18) from the Defense Meteorological Satellite Program (DMSP) provided SSMI/S UAS observations, together offering a unique source of operational temperature information in the upper stratosphere and lower mesosphere with excellent global coverage (Hoppel et al., 2013; McCormack et al., 2017). At present, only a single DMSP platform (F17) provides SSMI/S UAS observations and there are no plans to extend the UAS capability to any future missions.


NAVGEM-HA produces atmospheric data sets of winds, temperature, geopotential height, ozone, and water vapor by combining a hybrid 4-dimensional variational (or 4DVAR) DA solver with a global spectral atmospheric forecast model. The hybrid 4DVAR approach uses a linear combination of static (i.e., constant in time) model error covariance estimates and model error covariances estimated from 80-member ensembles of 6-hour forecasts that vary over time
(Kuhl et al., 2013). The present study uses a linear weighting factor of 0.5, meaning the static and time-dependent

model error covariances are equally weighted. Further details of the DA solver, including incorporation of middle atmosphere observation error and methods of bias correction between middle atmosphere satellite data sets, are provided in Kuhl et al. (2013) and Eckermann et al. (2018).

This intercomparison examines NAVGEM-HA zonal wind, temperature, and geopotential height fields produced with the T119L74 version of the system, where T119 refers to the triangular wavenumber truncation of the spectral forecast model and corresponds to a horizontal grid spacing of 1° in latitude and longitude, and L74 refers to 74 vertical model levels extending from the surface to the top pressure of 6 x $10^{-5}$ hPa. The NAVGEM-HA vertical coordinate is hybrid σ-p that is terrain following near the surface and transitions to isobaric above the 88 hPa level (approximately 17 km

altitude). The spacing of the model's vertical levels is ~2 km in the stratosphere, ~3 km in the mesosphere, and >4 km in the lower thermosphere. Strong horizontal diffusion is applied to the top two model levels (above ~ 100 km altitude) in order to prevent numerical instabilities resulting from, e.g., spurious wave reflection. The resulting analyses near the model top are heavily influenced by this imposed diffusion. Therefore, in this study we limit our focus to altitudes below 95 km geometric altitude, where previous validation studies (e.g., McCormack et al., 2017; Dhadly et al., 2018;

Stober et al., 2020) have shown NAVGEM-HA to produce reliable results. The NAVGEM-HA system produces analyses every 6 hours, and these fields are supplemented by 3-hourly forecast fields produced by the system as part of the 4DVAR framework, providing an effective 3-hourly sampling rate for the extraction of tidal signatures in the horizontal wind and temperature fields.

**2.2 MERRA-2**

MERRA-2 temperature, geopotential height, and zonal winds are used in this study (Gelaro et al., 2017). The 3-hourly fields on the native model grid ("3d_asm_Nv"; GMAO, 2015) provide the best time resolution available, with horizontal grid spacing of 0.625° longitude by 0.5° latitude on 72 vertical levels that extend from the Earth's surface to 0.01 hPa (~75 km). The vertical grid spacing is ~2 km in the upper stratosphere and lower mesosphere, increasing to

~5 km near 80 km altitude (see, e.g., Fujiwara et al., 2017). MERRA-2 assimilates a full range of ground based and satellite radiance observations, including the stratospheric channels of the available Advanced Microwave Sounding Unit (AMSU-A) instruments (McCarty et al., 2016). During the time period of interest here MERRA-2 assimilates Aura MLS temperatures from 5 to 0.02 hPa and ozone from 250 to 0.1 hPa to better constrain the dynamics in the upper stratosphere and mesosphere (Gelaro et al., 2017). The MERRA-2 model component contains a stratospheric

Quasi-Biennial Oscillation (QBO; Molod et al., 2015) and the MERRA-2 analysis QBO winds match well with the available radiosonde observations (Coy et al. 2016; Kawatani et al. 2016). While MERRA-2 has an equatorial semi-annual oscillation (SAO), Kawatani et al. (2020) has shown that reanalyses can differ in their representation of the SAO near the stratopause.

**2.3 JAGUAR-DAS**

JAGUAR is a comprehensive numerical model that extends from the Earth's surface to the lower thermosphere (~150 km). It is cooperatively developed by the Japan Agency for Marine-Earth Science and Technology (JAMSTEC), the Kyushu University, and the University of Tokyo based on the Model for Interdisciplinary Research on Climate (MIROC) and the Kyushu-GCM (Watanabe and Miyahara, 2009). A full set of physical parameterizations necessary to simulate altitudes from the surface to ~150 km is included, as described in Koshin et al. (2020). The JAGUAR model generates short-term forecasts that are used as background fields for the data assimilation system (JAGUAR-DAS), which employs a four-dimensional local ensemble transform Kalman filter (4D-LETKF) developed by Miyoshi and Yamane (2007). The forecast model has 124 vertical layers from the surface to ~150 km and a T42 horizontal resolution. The vertical grid spacing is 1 km in the 50-100 km altitude range. As the uppermost layers are taken as a sponge layer, only data below ~105 km altitude are usable for dynamical analysis. Following Koshin et al. (2020), the JAGUAR-DAS output used in the present study assimilates the standard National Centers for Environmental Prediction (NCEP) PREPBUFR dataset for the troposphere and lower stratosphere. For the stratosphere, mesosphere, and lower thermosphere, JAGUAR-DAS assimilates bias-corrected MLS temperature retrievals from 100–0.002 hPa (~16 km to 90 km altitude). The JAGUAR-DAS output used in the present study also includes three recent improvements: (1) introduction of incremental analysis update filtering to suppress generation of spurious waves; (2) a modified treatment of horizontal diffusion in the JAGUAR forecast model; (3) assimilation of SABER temperature retrievals from 40– 0.00014 hPa (~22 km to 110 km altitude) and the SSMI/S UAS microwave radiance measurements, described in section 2.1, from 10–0.01hPa (~30 km to 80 km). These improvements will be described in an upcoming study by Koshin et al. (2021, in review). Model error covariances were estimated from 50-member ensembles. The output from JAGUAR-DAS is 6-hourly and has horizontal grid spacing of 2.8125º in latitude and longitude.

## 2.4 WACCMX+DART

The background model in WACCMX+DART is WACCMX version 2.0 (Liu et al., 2018). WACCMX is an atmospheric component of the Community Earth System Model (CESM; Danabasoglu et al., 2020), and encompasses the whole atmosphere from the surface to the upper thermosphere (4.1 x $10^{-10}$ hPa, ~500 km to 700 km depending on solar activity conditions). WACCMX incorporates the chemical, dynamical, and physical processes from WACCM version 4 (Marsh et al., 2013) and the Community Atmosphere Model version 4 (Neale et al., 2014) in the lower-middle atmosphere. Additional T/I processes are incorporated in WACCMX, including major species diffusion, ionosphere transport of $O^{+}$, and self-consistent electrodynamics. The horizontal resolution of WACCMX is 2.5º in longitude and 1.9º in latitude. The vertical resolution ranges from ~1 km in the lower stratosphere to ~3 km in the upper mesosphere, and is ~4-5 km at higher altitudes. A detailed description of WACCMX version 2.0 can be found in Liu et al. (2018).

The data assimilation capability is implemented in WACCMX using the Data Assimilation Research Testbed (DART, Anderson et al., 2009) ensemble adjustment Kalman filter (Pedatella et al., 2014b, 2018). WACCMX+DART assimilates conventional meteorological observations (e.g., aircraft and radiosonde temperature and winds) and GPS

radio occultation refractivity in the troposphere-stratosphere, as well as Aura MLS and TIMED SABER temperature observations up to ~100 km altitude. To prevent spurious correlations, the observations are localized using a Gaspari-Cohn (Gaspari and Cohn, 1999) function with a half-width of 0.2 radians in the horizontal and 0.15 in $\ln(p_o/p)$ in the vertical, where p is pressure and $p_o$ is surface pressure. For the present study, WACCMX+DART simulations were performed using 40 ensemble members and a six-hour data assimilation cycle. Second order divergence damping was applied in order to stabilize the model as well as prevent large decreases in the $O/N_2$ ratio and electron density in the thermosphere and ionosphere (Pedatella et al., 2018). The second order divergence damping results in tidal amplitudes that are 50-100% too small. Pedatella et al. (2020) demonstrated that the tidal amplitudes can be improved by using hourly data assimilation cycling; however, the present study makes use of existing simulations that utilized a six-hour data assimilation cycle. The WACCMX+DART six-hourly analysis fields of zonal wind, temperature, and geopotential height are combined with short-term (1-5 hour) forecasts, yielding hourly output for analysis in the present study.

**2.5 Space-time analysis**

To quantify the various PW and tidal components in the high-altitude analyses, we employ the two-dimensional fast Fourier transform (2DFFT) method introduced by Hayashi (1971). Following McCormack et al. (2009), daily zonal means are subtracted from each hourly (WACCMX+DART), 3-hourly (MERRA-2 and NAVGEM-HA), or 6-hourly (JAGUAR-DAS) longitude-time field for a given month and then a cosine taper is applied to the first and last 10% of each record in time. The resulting power spectra describe the variance related to both eastward and westward propagating features as a function of frequency and zonal wavenumber. Individual components related to DW1, SW2, DE3, Q2DW, and Q5DW are isolated through the application of band-pass filters to the inverse 2DFFT (e.g., McCormack et al., 2009). The pass bands (described below) are determined by examining individual wavenumber-frequency spectra in middle atmosphere temperature anomalies from all four analyses over the DJF 2009-2010 period (not shown).

We also apply a continuous wavelet transform based on the S-transform method (Stockwell et al., 1996) to characterize the time variation of both migrating (DW1, SW2) and non-migrating (DE3) tidal components throughout the 2009-2010 winter. The S-transform has been used previously to examine the time behavior of the SW2 component in NAVGEM-HA wind fields during the 2009-2010 and 2012-2013 NH winters (McCormack et al., 2017), and we now extend this type of analysis to examine time variations related to DW1, SW2, and DE3 in the upper mesosphere from the NAVGEM-HA, JAGUAR-DAS, and WACCMX+DART data sets. Following the method described in McCormack et al. (2017), the S-transform produces estimates of wave amplitude as a function of both time and frequency. To evaluate the different tidal components with the S-transform, a one-dimensional FFT is first used to filter each data set to isolate the zonal wavenumber 1, 2, or 3 components, following Sassi et al. (2016). The S-transform is then applied to the horizontal wavenumber-filtered time series of temperature anomalies (time mean removed), and the resulting wave amplitudes at frequencies of 1 cycle per day (cpd) and 2 cpd are examined. Significance levels for these results are estimated following Torrance and Compo (1998), where we make use of the fact that the time-mean of the S-transform returns the exact Fourier spectrum. The time means of the S-transform results produce spectra that are

evaluated against a spectrum of a first-order autoregressive time series with the same variance as the input temperature time series, as described in Sassi et al. (2012). The 90% and 95% confidence values are constructed based on eq. (18) in Torrance and Compo (1998).

For this initial intercomparison, we examine all available output from these meteorological analyses over the altitude region from 20-120 km, with particular emphasis on the MLT region between ~50 km and 90 km altitude. Unless otherwise noted, all results are based on geometric altitude $Z$ computed using gridded geopotential height $H$ output by each system corrected for both altitude and latitude variations in gravitational acceleration following Lewis (2007):

$$Z = \frac{R_e(\phi)\,H}{\left(\frac{\gamma(\phi)}{\gamma_{45}}\right)R_e(\phi) - H}$$

where $H$ is the geopotential height in meters, $\phi$ is latitude in degrees, $\gamma_{45}$ is the surface gravitational acceleration at 45° latitude (9.80665 m s$^{-2}$), $R_e(\phi)$ is a latitude-dependent value of Earth's radius that corrects for the combined effect of gravitational and centrifugal forces, and the latitude-dependent gravitational acceleration $\gamma(\phi)$ on the surface of an ellipsoid of revolution is given by the expression

$$\gamma(\phi) = \gamma_e \left\{ \frac{1 + k_s\,sin^2(\phi)}{\sqrt{1 - e^2 sin^2(\phi)}} \right\}$$

using Somagliana's constant $k_s = 1.931853 \times 10^{-3}$, the Earth's eccentricity factor $e = 0.081819$, and the gravitational acceleration at the Equator $\gamma_e = 9.7803253359$ m s$^{-2}$.

## 3. Zonal mean results

To begin, we examine how each of the four high altitude meteorological analyses represent the latitude and altitude dependencies of zonal mean temperature and zonal mean zonal wind averaged over the DJF period 2009-2010. The zonal mean temperature distribution from 20 km to 120 km altitude plotted in Figure 2 reflects a balance between net radiative heating (driven primarily by stratospheric $O_3$ heating and mesospheric $CO_2$ cooling) and dynamically induced heating resulting from a thermally indirect (or residual) meridional circulation. This circulation is mainly produced by the cumulative effects of breaking PWs in the stratosphere and breaking gravity waves in the mesosphere. Similarly, the zonal mean zonal wind distributions plotted in Figure 3 from all four analysis systems also reflect this balance between radiative and dynamical drivers of the middle atmospheric circulation. Consequently, the zonal mean temperature and zonal wind distributions produced by each analysis system can depend not only on the number and quality of middle atmospheric observations being directly assimilated, but also by the physical parameterizations employed by the atmospheric model components to represent key processes (e.g., radiative heating and cooling, parameterization of sub-grid scale gravity wave drag). By characterizing similarities and differences in the zonal mean state among the four systems, we can begin to understand the relative roles that observations and model physics may play in producing these high-altitude meteorological data sets.

Between 20 km and 50 km altitude, the zonal mean temperature distributions among all four data sets are broadly similar, exhibiting temperatures below 210 K in the equatorial lower stratosphere near 20 km altitude, consistent with adiabatic cooling in the upward branch of the Brewer-Dobson circulation, as well as in the NH winter polar night region below ~30 km altitude. Each system produces temperature maxima of ~280 K near 50 km altitude at the South Pole related to peak ozone heating via absorption of solar UV radiation. The latitude structure of the stratopause varies somewhat among the different analyses, with JAGUAR-DAS exhibiting a local temperature maximum near 55 km altitude at the Equator, while WACCMX+DART exhibits little to no latitude variation in the altitude of the tropical temperature maximum. Near 80 km altitude, all four analyses are qualitatively similar, showing lower temperatures over the summer polar region arising from upward vertical motion, and higher temperatures over the winter polar region related to downward vertical motion. The upward and downward vertical motion over the poles in the mesosphere are both features associated with a global residual meridional circulation from summer to winter hemisphere driven by the effects of gravity wave drag; this circulation is represented by the broad arrow in Fig. 1. However, there are important quantitative differences among the DJF zonal mean temperature distributions, most notably in the tropics from 80 km to 100 km altitude, where WACCMX+DART produces temperatures that are >20 K warmer than corresponding temperatures produced by the NAVGEM-HA and JAGUAR-DAS systems. A warm bias at the tropical mesopause has been documented previously in free-running WACCM model simulations (e.g., Smith, 2012; Marsh et al., 2013; Harvey et al., 2019) but the cause is not yet fully understood. We also note that the summer polar temperature at 80 km altitude is ~20 K colder in WACCMX+DART compared to the other three data sets.

Also plotted in Figure 2 as heavy white contours are the corresponding temporal standard deviations of the zonal mean temperature during DJF from each analysis (see also supplemental Figure S1). All four analyses exhibit standard deviations exceeding 10 K at high northern latitudes, reflecting the relatively large amount of dynamical variability in the NH winter polar stratosphere associated with the SSW that occurred on 9 February. Large standard deviations are also noted at the summer polar mesopause, with NAVGEM-HA and JAGUAR-DAS values exceeding 10 K and WACCMX+DART values exceeding 20 K.

Figure 3 plots the DJF zonal mean zonal winds and temporal standard deviations from the four analyses (see also supplemental Figure S2). The general morphologies of the zonal mean zonal wind distributions in altitude and latitude are similar in all cases, exhibiting easterly (i.e., westward) flow in the summer hemisphere that tilts poleward with increasing altitude, and westerly (i.e., eastward) flow in the winter hemisphere that tilts equatorward with increasing altitude. However, there are significant quantitative differences that likely warrant future investigation, the most prominent being the stronger peak winds in WACCMX+DART. These differences are likely due to inaccurate specification of the background winds in the model (e.g., Marsh et al., 2013 see their Figure 1) and are most likely due to errors in the GW parameterizations. This work shows that these known wind biases are not fully corrected by the assimilation of stratospheric and mesospheric temperature observations. For instance, WACCMX+DART exhibits an easterly jet that exceeds 80 m s$^{-1}$ in the upper stratosphere and lower mesosphere (~50-60 km) between the Equator and 30°S latitude, whereas the analogous easterly jet in the other models is weaker and more variable (as indicated by

the standard deviation contours). Likewise, the westerly jet in the NH mid-latitude upper stratosphere and mesosphere (~50-80 km) is stronger in WACCMX+DART than in the other simulations. Differences are even more pronounced above 80 km. WACCMX+DART shows a westerly jet in the southern hemisphere (SH) that peaks near 35°-50°S and 100-105 km altitude, with wind speeds >70 m s$^{-1}$. Although both NAVGEM-HA and JAGUAR-DAS do exhibit westerly winds in the SH above 80 km, they are weaker than in WACCMX+DART in the respective regions of overlap (up to 95 km in NAVGEM-HA and 105 km in JAGUAR-DAS). Particularly notable is that even though JAGUAR-DAS extends to ~105 km, the SH westerly winds at this altitude only reach ~25 m s$^{-1}$, more than 40 m s$^{-1}$ slower than in WACCMX+DART. An exception to the stronger peak winds in WACCMX+DART is evident in the NH lower thermosphere (~90-105 km altitude) from 0°-50°N latitude, where JAGUAR-DAS shows a strong easterly jet (>40 m s$^{-1}$ near 30°N); but WACCMX+DART easterlies in the NH lower thermosphere are weaker and shifted to higher latitudes. Finally, in the tropical lower stratosphere, NAVGEM-HA, MERRA-2, and JAGUAR-DAS capture the alternating easterly and westerly flow related to the QBO, while WACCMX+DART shows easterly flow throughout the tropical stratosphere.

Examining the standard deviations in the DJF zonal mean winds in Figure 3, we see that NAVGEM-HA, MERRA-2, and JAGUAR-DAS all exhibit similar variability along the equatorward flank of the summer easterly jet, but this variability is not present in WACCMX+DART. In the NH winter stratosphere, all four data sets exhibit similar variability associated with the stratospheric polar night jet. Above 80 km, the major difference is the large variability in WACCMX+DART zonal mean zonal winds in the lower thermosphere between 30° and 50°S, coincident with the strong westerly jet.

The results in Figs. 2 and 3 show that the largest differences occur above 80 km, where effects of gravity wave drag play an important role in determining the climatological zonal mean distributions of temperature and zonal wind in the middle atmosphere. Specific features such as the latitude and altitude dependences of the mesospheric summer easterly jet and the cold summer polar mesopause are known to be sensitive to the effects of gravity wave breaking and subsequent deposition of heat and momentum into the background (zonal mean) state (e.g., Fritts and Alexander, 2003). Some of the largest differences among the standard deviations in both zonal mean temperature and zonal mean zonal wind plotted in Figs. 2 and 3 occur in the vicinity of these features, suggesting that differences in the treatment of gravity wave drag may be an important factor for explaining the large differences among the analyses above 80 km. Indeed, Pedatella et al. (2014a) showed that gravity wave drag differences among models is related to differences in the background winds. The cause of the temperature and zonal wind differences presented here requires further investigation that is beyond the scope of this initial intercomparison study.

To further examine the differences in zonal mean temperature and zonal wind distributions among the four analyses, Figure 4 plots the latitude distribution of zonal mean temperature (left column) and zonal mean zonal wind (right column) at 80 km (top) and 50 km (bottom) averaged over January 2010, when the variability of the NH winter zonal mean winds and temperatures in the mesosphere were largest due to the occurrence of the SSW. To evaluate differences

in the intrinsic variability in these quantities during NH winter, Figure 4 also shows standard deviations about the January mean as a function of latitude. At 50 km altitude (Fig. 4, bottom row), we find that the zonal mean temperature and zonal wind values among the four analyses are in very good agreement in the SH (summer) extratropics, where the day-to-day variability throughout the month is relatively small. Near the Equator, the temperatures at 50 km differ by 8-10 K, with MERRA-2 and NAVGEM-HA tending to be warmer and WACCMX+DART tending to be cooler. However, there is a very large spread (~80-100 m s$^{-1}$) among the January mean zonal winds at 50 km within the tropics, with NAVGEM-HA exhibiting weak westerly winds at the Equator and WACCMX+DART exhibiting strong easterly winds. These differences in equatorial zonal mean zonal wind at 50 km among the four analyses are much larger than the day-to-day variability indicated by the corresponding standard deviation values, suggesting a systematic bias could be present among these data sets. At NH extratropical latitudes, all four analyses produce similar mean values, and the spread among the mean results is much smaller than the standard deviations. The large standard deviations in the extratropical NH (winter) at 50 km reflect the high degree of day-to-day variability due to strong PW forcing in late January that resulted in a major SSW on 9 February.

In contrast to the results at 50 km, at 80 km altitude (Fig. 4, top row), we find significant differences in both zonal mean temperature and zonal mean zonal wind values throughout the extratropical SH. Most notably, WACCMX+DART exhibits temperatures up to ~20 K cooler near 70$^{o}$S and weak westerly winds near 50$^{o}$S, in contrast to strong easterlies in MERRA-2, NAVGEM-HA and JAGUAR-DAS. Similar to the results at 50 km, the equatorial zonal mean zonal winds at 80 km also exhibit considerable spread, and these differences are larger than the temporal standard deviation during January 2010. The large differences in equatorial zonal winds at both 50 km and 80 km highlight the challenge of producing wind analyses in a region where geostrophic balance constraints used by DA systems (see, e.g., Eckermann et al., 2018, their Figure 4) to relate wind information to the satellite-based middle atmosphere temperature observations (e.g., MLS, SABER) begin to break down.

In addition to the monthly and seasonally averaged results presented in Figures 2-4, comparisons of the daily variability in zonal mean temperatures and zonal winds are of interest because the 2009-2010 NH winter was so dynamically active. The major SSW that took place on 9 February 2010 was preceded by a reversal in mesospheric flow from westerly to easterly beginning on 27 January, which then descended to the stratosphere (McCormack et al., 2017). This mesospheric wind reversal effectively filters out upward propagating gravity waves with westward phase speeds through the formation of a critical line, thereby dramatically reducing dynamical heating via gravity wave breaking in the NH polar mesosphere. The result is the well-documented "sudden mesospheric cooling" that accompanies most SSW events (e.g., Matsuno, 1971; Labitzke, 1972; Siskind et al., 2010; Eswaraiah et al., 2017). It has been suggested that the abrupt changes in NH (winter) polar gravity wave breaking can have consequences for SH (summer) polar mesopause temperatures through changes in the pole-to-pole meridional residual circulation produced by subsequent modulation of the gravity wave drag in both the winter and summer mesosphere (e.g., Karlsson and Becker, 2016; Laskar et al., 2019; Zülicke et al., 2018). The combined effects of these SSW-related changes in mesospheric gravity wave drag produce an anomalous residual circulation with weaker upwelling in the summer polar mesopause region,

and thus warmer temperatures in this region due to a reduction in adiabatic cooling. Alternatively, several case studies based on high-altitude meteorological analyses suggest that changes in mesospheric Q2DW activity may play a role in interhemispheric coupling (e.g., Siskind and McCormack, 2014; France et al., 2018; Lieberman et al., 2021). An additional mechanism was discussed in Smith et al. (2020), where changes in summer polar mesopause temperatures are a response to changes in the residual meridional circulation, with no direct role for wave activity in the summer hemisphere.

The relationship between winter mesospheric cooling and summer polar mesopause warming for the 2009-2010 NH winter period is examined in Figure 5, which plots the time behaviour of daily averaged zonal mean temperatures at 80°S (left column) and 80°N (right column) from 1 December 2009 to 31 March 2010. There are two key dates highlighted in each panel. The left vertical red line denotes 27 January 2010, the first day of sustained (>5 days) mesospheric easterly winds at 60°N (McCormack et al., 2017). Easterly winds in the upper stratosphere have been shown to be an effective proxy to explore mesospheric and lower thermospheric effects following SSWs (Jones et al., 2018; Limpasuvan et al. 2016; Stray et al., 2015; Tweedy et al., 2013). The right red vertical line indicates 9 February 2010, the onset of easterly winds in the stratosphere (Butler et al., 2017). These two dates are highlighted throughout the paper to denote the disturbed stratospheric and mesospheric time period. At 80°N (right column), all four analyses agree with respect to the timing of the SSW, and the three analyses that extend above 80 km altitude also show similar timing of the mesospheric cooling. We note that the winter mesopause is at ~90-95 km in NAVGEM-HA but is near 100 km in both JAGUAR-DAS and WACCMX+DART. At 80°S (left column) the main differences are in the minimum temperature values from 85 to 95 km altitude, where the NAVGEM-HA minimum value is ~140 K, the JAGUAR-DAS minimum value is ~130K, and the WACCMX+DART minimum value is ~120 K. The lower altitude and warmer temperatures at the high southern latitudes in NAVGEM-HA may be a consequence of the lower model top. There are also differences in the seasonal evolution of the cold summer polar mesopause, most notably the downward progression of the temperature minimum in WACCMX+DART during January and February, which is not seen in either NAVGEM-HA or JAGUAR-DAS. None of the high-altitude analyses show a clear relationship between the onset of the mesospheric cooling at 80°N and an increase in summer polar mesopause temperatures at 80°S that would indicate a direct interhemispheric coupling (IHC) mechanism as described above, although we note that previous studies found the temperature response in the summer mesopause region to be relatively small, ~2-5 K (e.g., Karlsson et al., 2009a; deWit et al., 2015). Further examination of output from these analyses for other SSW cases in conjunction with modeling studies is needed to fully explore possible links between summer polar mesopause warmings and middle atmospheric variability in NH winter.

Similar to the zonal mean temperature results in Fig. 5, all four analyses exhibit similar temporal behavior in the zonal mean zonal winds at 60°N (Fig. 6, right column) during the 2009-2010 winter period up to ~70 km altitude, capturing both the sudden reversal of mesospheric winds in late January and the downward descent of easterly zonal winds into the stratosphere. Above 70 km altitude, the main differences are the presence of weak westerly flow in NAVGEM-HA, JAGUAR-DAS, and MERRA-2 (up to 80 km) whereas WACCMX+DART produces easterly flow above 70 km

with maximum values exceeding -30 m s$^{-1}$ from 80 to 100 km altitude. At 60$^o$S (Fig. 6, left column), all four analyses show an easterly jet centered near 75 km altitude in December 2009. Above this level, WACCMX+DART shows much larger vertical wind shear compared to NAVGEM-HA and JAGUAR-DAS and a rapid transition to strong westerly flow exceeding 60 m s$^{-1}$ in the lower thermosphere. Since the deceleration and reversal of the easterly summer mesospheric jet is related to strong eastward gravity wave drag, differing treatments of gravity wave drag among the various systems, most notably in WACCMX+DART, may be responsible for the differences in the vertical structure of the easterly summer jet at 60$^o$S in Fig. 6. Further investigation of this would require detailed momentum budget studies using specific output data (e.g., wind tendencies due to parameterized wave drag) that are not available for the present study. Making this data part of standard meteorological output fields would facilitate future investigations into the specific role that gravity wave drag plays in explaining these differences among the mesospheric zonal wind analyses.

To further explore the global response of middle atmospheric zonal mean zonal winds and temperatures to the occurrence of the SSW and mesospheric cooling in the NH winter of 2009-2010, we next examine the latitude/time distributions of zonal mean temperature and zonal mean zonal wind for three altitudes (50 km, 70 km and 90 km) in Figures 7 and 8, respectively, from the four analyses. Overall, we find good qualitative and quantitative agreement among the zonal mean temperatures at 50 km (Fig. 7, bottom row). We note that NAVGEM-HA and MERRA-2, which assimilate MLS stratospheric O$_3$ profiles, exhibit slightly lower peak temperatures at the South Pole compared to JAGUAR-DAS and WACCMX+DART, which do not assimilate stratospheric O$_3$ observations. It would be of interest for future work to examine how differences in the assimilation of radiatively active chemical constituents such as O$_3$ and H$_2$O impact the agreement among different middle atmospheric meteorological analyses. At 70 km altitude (Fig. 7, middle row), there is generally good qualitative agreement among the four analyses. Notable quantitative differences are the comparatively warmer temperatures in the equatorial region, and the comparatively colder temperatures from 50$^o$-90$^o$S, during late February and March in WACCMX+DART. At 90 km altitude (Fig. 7, top row), we again find generally consistent qualitative behavior, but with some important quantitative differences. Specifically, NAVGEM-HA shows a pronounced mesospheric cooling in the NH extratropics in mid-December that is not present in the JAGUAR-DAS or WACCMX+DART results. JAGUAR-DAS equatorial temperatures are 10-15 K colder than NAVGEM-HA or WACCMX+DART. At the South Pole, WACCMX+DART temperatures are 20-30 K colder than NAVGEM-HA or JAGUAR-DAS. While all three high-altitude analyses show the mesospheric cooling prior to the major SSW in early February 2010, only NAVGEM-HA and WACCMX+DART indicate a related warm anomaly in the equatorial regions.

The latitude-time distributions of zonal mean zonal wind, shown in Figure 8, also generally show good qualitative agreement among the four analyses regarding the timing of the wind reversals in the NH extratropics related to the SSW and mesospheric cooling seen in Fig. 7. Notable differences in the behavior of the zonal mean zonal winds include: the very strong and persistent easterly flow in the equatorial regions at 50 km altitude (Fig. 8, bottom row) seen in WACCMX+DART; the emergence of tropical easterly flow in late February and March at 70 km altitude (Fig.

8, middle row) in NAVGEM-HA and the split summer easterly jet in the SH seen in WACCMX+DART; easterly winds over the Equator at 90 km altitude (Fig. 8, top row) in the JAGUAR-DAS results and the strong westerly flow in the WACCMX+DART results near 40°S, which was also noted in the discussion of DJF average results (Fig. 3, bottom right panel). These zonal wind differences in the upper stratosphere and mesosphere are likely attributed to differences in the treatment of gravity wave drag in each system, though specific origins require further investigation, as noted above. Users of these high-altitude meteorological analyses should be aware that these differences in the zonal mean zonal winds imply that the choice of meteorological inputs may impact the results of nudged whole atmosphere simulations.

Next we explore global temperature variations during the two weeks preceding the major SSW event. Figure 9 shows latitude-altitude plots of the correlation coefficient between daily mean temperature variations at 80°N and 30 km and corresponding temperature variations at other latitudes and altitudes during 27 January to 9 February 2010 in the four analyses. As expected, all four systems show positive correlations (warming) in the NH polar stratosphere, evidence that they all simulate the SSW event. Likewise, all four systems show negative correlations (cooling) in the NH polar mesosphere; this demonstrates that mesospheric cooling is also reliably captured in all systems. Similar connections between the SSW and polar mesospheric temperatures have been noted in previous observational studies using MLS temperature data (e.g., Zülicke et al., 2018). However, Fig. 9 indicates that there are also consistent correlation coefficient patterns that extend into the deep tropics and into the SH among the four systems. All four systems show vertically alternating negative and positive correlation regions in the tropics and subtropics of both hemispheres. All four systems show negative correlations (cooling) in the SH polar stratosphere and lower mesosphere and positive correlations (warming) poleward of 40°S between ~75 and 95 km, consistent with inter-hemispheric coupling relationships reported by Karlsson et al. (2009b). The agreement in temperature variability among the systems in the NH polar stratosphere and mesosphere is expected. However, the agreement in temperature variations among the systems in the tropics and in the summer hemisphere, even extending into the upper mesosphere, demonstrates that the four analyses capture similar temporal behavior globally, despite the mean differences shown earlier.

To examine the range in zonal mean temperatures and zonal winds, Figures 10 and 11 plot the standard deviations in the daily mean values of each quantity among the four analyses (three at 90 km where MERRA-2 is unavailable). Fig. 10 shows that all the analyses are in fairly good quantitative agreement with regards to temperature at 50 km and 70 km altitude, but deviations of 10 K or more are common at 90 km, with the largest disagreement occurring at the South Pole at the end of summer. Similarly, Fig. 11 shows that zonal wind deviations among the data sets are generally 5 m s$^{-1}$ or less outside of the equatorial regions at 50 and 70 km, but larger deviations in excess of 20 m s$^{-1}$ emerge at 90 km both in the tropics and near 50° latitude in both hemispheres. Overall, the largest zonal mean zonal wind deviations (>35 m s$^{-1}$) occur not at the higher altitudes, but at 50 km altitude during February and March 2010 (Fig. 11, bottom panel). The results in Fig. 11 indicate that these high-altitude analyses do not yet produce a consistent representation of the semi-annual oscillation (SAO) in zonal mean zonal winds in the equatorial middle atmosphere (Kawatani et al., 2020). The SAO is a basic climatological feature of the middle atmospheric circulation that impacts the propagation of

gravity waves and tides into the mesosphere and lower thermosphere (e.g., Garcia et al., 1997). Consequently, this is an issue that will need to be addressed as these high-altitude data assimilation systems evolve.

## 4. Planetary Wave and Tide Results

In addition to zonal mean quantities, these four middle atmosphere meteorological analyses also provide valuable information on zonal variations in temperature and winds related to planetary scale waves and tides, which earlier studies based on MLS (e.g., Forbes and Wu, 2006) and SABER (e.g., Garcia et al., 2005; Zhang et al., 2006) temperature observations found to be prevalent throughout the MLT. Since each of the four analyses examined here assimilate either MLS data, SABER data, or a combination of the two, this section examines how these features are captured in each of the reanalyses. To begin, Figure 12 plots longitude-time variations in daily mean temperature at 60ºN and 70 km altitude from 1 December 2009 to 31 March 2010. At this altitude, there is good agreement in the zonal variations in temperature among the four analyses, which all show a strong quasi-stationary zonal wavenumber 1 during December 2009 and January 2010, which then abruptly shifts to a slowly propagating westward wavenumber 1 feature in early February that persists through March. The timing of this shift appears to coincide with the reversal of mesospheric winds on 27 January, two weeks before the major SSW, as shown in Figure 6. We note that the quasi-stationary and traveling PW amplitudes are larger in WACCMX+DART relative to the NAVGEM-HA, MERRA-2, and JAGUAR-DAS results. Abrupt shifts in quasi-stationary planetary wave 1 in the Northern high latitude winter mesosphere related to SSWs have been documented in numerous studies (e.g., Smith, 2003; Manney et al., 2008; Siskind et al., 2010; Chandran et al., 2013; Koushik et al., 2020), and are linked to highly episodic sources of barotropic/baroclinic instability at NH middle and high latitudes within the upper stratosphere and mesosphere (Sassi and Liu, 2014). Future studies comparing the relative roles of resolved vs. parameterized gravity wave forcing of the mesospheric circulation, as well as the representation of baroclinic/barotropic instabilities, within the four analyses could lend insight into the origins of the differences in Fig. 12, and would help to improve our understanding of this phenomenon as it relates to changes in the state of the T/I system in connection to SSWs.

In the remainder of this section, we present results from space-time analysis of the four analyses related to the Q5DW, Q2DW, DW1, SW2, and DE3 features. Recognizing that many other planetary wave and tidal features (e.g., Forbes et al., 2008; Sassi et al., 2012) are also important for producing T/I variability related to meteorological forcing from the middle atmosphere (McDonald et al., 2018), the present study is not meant to be an all-inclusive assessment of every feature, but rather is meant to provide an initial extension of the intercomparison study by Harvey et al. (2021) to include the mesosphere and lower thermosphere.

We begin with an examination of the Q5DW, which consists of a westward propagating zonal wavenumber 1 disturbance related to the first hemispherically symmetric normal (Rossby) mode. As shown in Harvey et al. (2021), the middle atmospheric Q5DW can manifest in two forms. First, as a hemispherically symmetric feature related to latent heat release in the tropical upper troposphere (Salby, 1981; Miyoshi and Hirooka, 2003) peaking between 30º and 50º latitude in the summer hemisphere. Second, as a high latitude wintertime feature related to growth through

baroclinic/barotropic instability, leading to what is commonly referred to as the 6.5-day wave in the mesosphere and lower thermosphere (Talaat et al., 2001; Lieberman et al., 2003; Forbes and Zhang, 2017). Given the complex dynamical interactions that give rise to the Q5DW, capturing this feature is a good test for middle atmospheric meteorological analyses. Figure 13 plots altitude and latitude dependences of the Q5DW amplitude in temperature during January 2010 extracted from the four analyses using the 2DFFT method described in section 2, using a bandpass for westward zonal wavenumber 1 and 0.16 – 0.24 cpd (periods of 4.25-6 days). In all four analyses, the dominant Q5DW pattern is the high-latitude winter feature with peak amplitudes of 2-3 K between $60^{\circ}$N and $80^{\circ}$ N latitude. These amplitudes are consistent with the 5-day Rossby normal mode variation of 2.5-3.5 K derived from SABER temperature observations in the study by Garcia et al. (2005) for the March-May 2002 period; they are also consistent with quasi-6-day wave amplitudes at high northern latitudes in January reported by Forbes and Zhang (2017) using 14 years of SABER temperatures. The main difference in the Q5DW amplitudes among the data sets is its vertical extent. Both NAVGEM-HA and WACCMX+DART exhibit Q5DW amplitudes of 1-2 K at high Northern latitudes above 80 km, whereas the corresponding Q5DW amplitudes in JAGUAR-DAS are limited to below 80 km altitude. The three analyses extending above 80 km altitude also indicate weak (0.5-1 K) Q5DW amplitudes in the SH (summer) extratropics that may be related to convective latent heat release.

Similar to the Q5DW, the Q2DW is a well-documented feature of upper stratospheric and mesospheric dynamics (e.g., Coy, 1979; Harris, 1994; Limpasuvan & Wu, 2003; Garcia et al., 2005; Pancheva, 2006; Lilienthal and Jacobi, 2015; Kumar et al., 2018). The Q2DW consists primarily of a westward propagating zonal wave number 3, although westward wave number 2 and 4 components are also present in both satellite observations and high-altitude meteorological analyses (e.g., McCormack et al., 2009; Tunbridge et al., 2011; Gu et al., 2013; McCormack et al., 2014). In the mesosphere, the Q2DW originates primarily from regions of baroclinic instability in the easterly mesospheric summer jet (Plumb, 1983; Pfister, 1985) that form in part by the effects of gravity wave drag (e.g., Ern et al., 2013; Sato et al., 2018). In the tropical upper stratosphere, the Q2DW can originate from regions of barotropic instability (Burks & Leovy, 1986) related to inertial instability resulting from unusually strong PW activity in the winter hemisphere (e.g., Orsolini et al., 1997; McCormack et al., 2009; Lieberman et al., 2021). Both observational and modeling studies have indicated that the Q2DW, often through interaction with tides, is a significant source of day-to-day variability in the dynamics and composition of the thermosphere and ionosphere (e.g., Chang et al., 2011; Yue et al., 2012; Chang et al., 2014). It is, therefore, important that meteorological analyses used to constrain whole atmosphere simulations accurately capture the Q2DW.

Figure 14 plots altitude and latitude dependences of the January monthly mean Q2DW amplitude in temperature extracted from the four analyses using a bandpass for zonal wavenumber 3 and westward frequencies between 0.45 and 0.6 cpd (periods of 1.6-2.2 days). Below 80 km altitude, all four analyses show largest Q2DW amplitudes in the SH along the equatorward flank of the summer easterly jet (see Fig. 3), coinciding with the region where the standard deviations in zonal mean zonal wind are largest in SH summer (e.g., Fig. 3 and Fig. S2). This spatial structure is broadly consistent with results from earlier studies based on MLS (e.g., Limpasuvan and Wu, 2003) and SABER (e.g., Gu et

al., 2013) temperature observations. Peak amplitudes range between 2-3 K in three of the analyses (NAVGEM-HA, MERRA-2, and JAGUAR-DAS), but are ~1 K in WACCMX+DART. Between 80 and 100 km altitude, both JAGUAR-DAS and WACCMX+DART indicate Q2DW amplitudes of 1-2 K between 30º-60ºS. There is also evidence of a small 1-2 K Q2DW feature in the NH between approximately 20º-30ºN latitude in NAVGEM-HA and JAGUAR-DAS. The quantitative differences in Q2DW amplitudes among the four analyses are likely related to the differences in the structure of the SH summer easterly jet in the upper stratosphere and mesosphere seen in Figs. 3 and 8. Specifically, WACCMX+DART exhibits much stronger easterly flow and less westerly wind shear in the subtropical stratopause region as compared to the other three analyses, and this may result in an environment that does not promote the growth of the Q2DW in the WACCMX+DART system to the extent seen in NAVGEM-HA, MERRA-2, or JAGUAR-DAS. We note that WACCMX+DART, NAVGEM-HA, and JAGUAR-DAS assimilate both MLS and SABER temperatures, whereas MERRA-2 assimilates MLS temperatures. This suggests that differences in the models themselves, rather than the data inputs, may explain the different Q2DW results in Fig. 14.

Next, we examine MLT tidal features in the four analyses. The latitude and altitude dependences of the January 2010 mean diurnal (DW1) and semi-diurnal (SW2) migrating solar tidal amplitudes are plotted in Figures 15 and 16, respectively. Monthly mean DW1 amplitudes in temperature were determined using a bandpass filter for zonal wave number 1 and westward frequencies between 0.9 – 1.1 cpd. In the stratosphere, all four analyses show similar DW1 signatures centered on midlatitudes in both hemispheres, similar to those reported by Sakazaki et al. (2012). Between 50 km and 80 km altitude, all four analyses also exhibit similar maxima in DW1 near the Equator with values of ~1-3 K, similar to results published previously (e.g., Forbes and Wu, 2006). Between 80 and 100 km altitude, NAVGEM-HA, JAGUAR-DAS, and WACCMX+DART exhibit maxima in the equatorial regions as well as secondary maxima between 30º and 50º latitude in each hemisphere. The main difference between the DW1 amplitudes among the three analyses extending above 80 km are the magnitude and vertical location of the equatorial maximum. The NAVGEM-HA DW1 amplitude peaks at ~7 K at 80-90 km altitude, while in MERRA-2 the peak DW1 amplitude of ~4K is near 75 km, in JAGUAR-DAS the peak DW1 amplitude of ~9K is located between 95 km and 100 km altitude, and in WACCMX+DART the peak DW1 amplitude of ~10K occurs near 110 km altitude. The range of altitudes for maximum DW1 amplitudes seen in these four analyses agrees with SABER observations (Zhang et al., 2006). For MERRA-2 and possibly NAVGEM-HA, the analysis system upper boundaries are low enough that artificially damping of DW1 may occur. In JAGUAR-DAS, DW1 is dissipated above ~100 km due to the model diffusion exponentially increasing with height to mimic molecular diffusion. The differences in DW1 structure at/above 100 km between JAGUAR-DAS and WACCMX+DART are likely due to the large differences in background zonal mean zonal wind (Fig. 3).

Figure 16 plots January 2010 mean amplitudes of SW2 in temperature obtained using a bandpass filter for zonal wave number 2 and westward frequencies between 1.95 cpd and 2.05 cpd for NAVGEM-HA, MERRA-2, and WACCMX+DART. For JAGUAR-DAS, the bandpass filter cuts off at 2.0 cpd, which is the Nyquist frequency for the 6-hourly output. Perhaps because of the wide range (1 h to 6 h) of output frequency among the four data sets, the derived amplitudes of the higher frequency SW2 vary considerably. There are some qualitative similarities in the

latitude structure of the SW2 amplitudes between 80 km and 100 km altitude, where NAVGEM-HA, JAGUAR-DAS, and WACCMX+DART all exhibit three peaks near 25°S, 15°N, and 40°N latitude. Near 40°N latitude, the peak SW2 amplitude in NAVGEM-HA of ~5 K occurs below 95 km, whereas JAGUAR-DAS and WACCMX+DART indicate peak SW amplitudes ranging from 6 to 8 K occur above 100 km altitude. This suggests that NAVGEM-HA may be missing key features of the SW2 due to its lower model top. Between 100 and 120 km altitude, WACCMX+DART indicates SW2 amplitudes of >20 K from 20°S to 40°S latitude.

In addition to migrating tides, nonmigrating tides are known to also impact T/I variability. One prominent nonmigrating feature is the eastward propagating diurnal zonal wave number 3 (DE3), that has been shown to play a role in establishing pronounced zonal variations in ionospheric total electron content (e.g., Immel et al., 2006; Hagan et al., 2007; McDonald et al., 2018). Variations in DE3 amplitude in relation to SSWs have been noted (Maute et al., 2014), with non-linear wave-wave interactions within the mesosphere playing an important role in DE3 growth (Lieberman et al., 2015; Sassi et al., 2021). Figure 17 plots the altitude and latitude dependencies of monthly mean DE3 amplitudes for January 2010 obtained using a bandpass filter for zonal wave number 3 and eastward frequencies between 0.9 cpd and 1.1 cpd. Overall, DE3 is a feature of the mesosphere and lower thermosphere, although there is some evidence for very small (~1 K) DE3 amplitudes near the stratopause in MERRA-2 (at ~35°S) and JAGUAR-DAS (at ~5°S). Between 60 km and 80 km altitude, the distribution of DE3 amplitudes in NAVGEM-HA, MERRA-2, and JAGUAR-DAS are roughly similar, showing amplitudes of ~2K near 40°-50°S and 10°-20°N latitude. JAGUAR-DAS also indicates DE3 amplitudes of ~2 K in the northern extratropics near 80 km. Above 80 km, NAVGEM-HA and JAGUAR-DAS show peak DE3 amplitudes of 3-4 K in the SH subtropics. The DE3 signature in WACCMX+DART is notably smaller than the other analyses, showing a single peak of ~3 K near the Equator between 100 km and 120 km altitude.

For purposes of constraining whole atmospheric model experiments, perhaps more important than the monthly mean amplitudes of the tides is the day-to-day tidal variability in the mesosphere and lower thermosphere captured by each of the three high-altitude analyses: NAVGEM-HA, JAGUAR-DAS, and WACCMX+DART. There is now substantial evidence that circulation changes throughout the stratosphere and mesosphere related to SSWs can modulate the solar migrating tides (Pedatella and Forbes, 2010; Lima et al., 2012; Pedatella and Liu., 2013). Typically, the amplitude of DW1 is seen to decrease in the days leading up to a SSW, followed by a pronounced increase in the amplitude of the SW2 for several days or weeks following the onset of the SSW (e.g., Pedatella and Liu, 2013; Limpasuvan et al., 2016; McCormack et al., 2017). The origins of the tidal modulation by SSWs are still under investigation, but possible causes may include transport-induced changes in the distribution of ozone heating in the equatorial upper stratosphere (Goncharenko et al., 2012; Siddiqui et al., 2019) and variations in zonal mean zonal winds that affect the upward propagation of the tides (McLandress, 2002; Sassi et al., 2013).

Figures 18, 19, and 20 show the time variations in the amplitudes of diurnal wave number 1, semidiurnal wave number 2, and diurnal wave number 3 in temperature at 90 km as a function of latitude throughout the course of the 2010 SSW and subsequent polar vortex recovery phase. These time variations are obtained using the FFT and S-transform methods

described in Section 2. We note that the S-transform method by itself does not distinguish between eastward and westward propagating features. However, based on examination of individual 2DFFT spectra (not shown), we find that

the dominant spectral features associated with diurnal wave 1, semidiurnal wave 2, and diurnal wave 3 in the temperature fields at this level correspond to DW1, SW2, and DE3, respectively. To avoid edge effects commonly associated with wavelet methods, results for the first and last three days in the time period are not plotted in Figs. 18, 19, and 20.

The latitude-time variations of DW1 temperature amplitudes at 90 km altitude from 1 January to 31 March 2010 from the three high altitude analyses in Figure 18 all show qualitatively consistent behavior, most notably a reduction in equatorial amplitudes in early February and a broad increase in amplitudes throughout the topics and subtropics approaching equinox conditions in March, when climatological DW1 temperature amplitudes are largest. During the January – March 2010 period, peak WACCMX+DART diurnal wave 1 amplitudes are roughly half as large as values

in NAVGEM-HA and JAGUAR-DAS. We note that the DW1 results from all 3 analyses plotted in Fig. 19 exceed the 95% confidence level at most latitudes.

For SW2 (Fig. 19), the peak WACCMX+DART amplitudes at 90 km are also generally less than peak values in the NAVGEM-HA or JAGUAR-DAS results. However, we note that in early February, all three high altitude data sets

indicate similar increases in the semidiurnal wave 2 amplitude of ~8-10 K near 10$^{\circ}$N latitude that exceed the 95% confidence levels. The NAVGEM-HA results show amplitudes of ~8 K in SW2 near 40$^{\circ}$-50$^{\circ}$N throughout February that are not present in WACCMX+DART or JAGUAR-DAS results. In addition, the JAGUAR-DAS results indicate numerous short-lived large amplitude features at high latitudes not seen in NAVGEM-HA or WACCMX+DART. Given the 6-hourly sampling of JAGUAR-DAS, these high latitude maxima may be an artifact produced by aliasing

of higher frequency variations, since the 2.0 cpd semidiurnal frequency corresponds to the Nyquist limit for JAGUAR-DAS output.

The time variations in DE3 at 90 km (Fig. 20) show some qualitative similarities among the three analyses, most notably a 30–40-day modulation of peak amplitudes throughout the 50$^{\circ}$S-50$^{\circ}$N latitude region that exceeds the 95%

confidence estimate. Given the relationship of the nonmigrating DE3 tide to convective sources (e.g., Forbes et al., 2008), these low-frequency variations could be a manifestation of intra-seasonal modes such as the Madden-Julian Oscillation, which has been shown to have a signature in the T/I system on timescales longer than 30 days (e.g., Sassi at al., 2019). As with the diurnal wave 1 and semidiurnal wave 2 results, the amplitudes of the diurnal wave 3 temperature variations at 90 km throughout the January – March 2010 period in Fig. 20 derived from

WACCMX+DART are generally a factor of 2 smaller than amplitudes derived from the NAVGEM-HA or JAGUAR-DAS temperature data sets.

To summarize the differences among NAVGEM-HA, JAGUAR-DAS, and WACCMX+DART associated with each of the zonal wave number – frequency pairs plotted in Figs. 18-20 prior to the major SSW of 2010, Figure 21 plots

latitude distributions of the mean amplitude and ±1 standard deviation in the temperature amplitudes for the period 27 January – 9 February derived from the S-transform analysis. For diurnal wave 1 (Fig. 21a), all three analyses show largest amplitudes near the Equator; NAVGEM-HA and JAGUAR-DAS peak values are both ~7K, while the WACCMX+DART peak value is ~4K. For semidiurnal wave 2 (Fig. 21b), all three analyses exhibit similar peak values from 5-15$^o$N with maximum amplitudes from 6-8K. Between 30$^o$N and 40$^o$N, both NAVGEM-HA and JAGUAR-DAS show a secondary peak in SW2 amplitude of ~5K, while corresponding WACCMX+DART values are ~3K. In addition, JAGUAR-DAS results show a secondary peak in SW2 amplitude between 10$^o$S and 20$^o$S latitude; this peak is smaller amplitude in NAVGEM-HA and is shifted poleward (to ~20$^o$S-40$^o$S) in WACCMX+DART. Results from JAGUAR-DAS show larger SW2 amplitudes at high latitudes (80$^o$S-90$^o$S and 60$^o$N-90$^o$N) not found in either NAVGEM-HA or WACCMX+DART. For diurnal wave 3 (Fig. 21c), both NAVGEM-HA and JAGUAR-DAS indicate peak DE3 values of ~4K between the Equator and 20$^o$S latitude, while WACCMX+DART shows no indication of a distinct DE3 signal.

Overall, the results in Figure 21 suggest that while there is general qualitative agreement in the latitude structure of DW1, SW2, and DE3 among the three meteorological analyses extending to 90 km altitude, there are important quantitative differences. These differences are likely related to the details of each assimilation system regarding the type of observations being assimilated, the type of atmospheric model employed, and differences in the temporal and spatial resolutions of each system. For example, is it possible that the 6-hourly output of JAGUAR-DAS, which is at the Nyquist frequency for SW2, may result in some aliasing of other signals; this could potentially explain some of the high-latitude SW2 amplitudes seen in JAGUAR-DAS (Fig. 21b) but not in either NAVGEM-HA or WACCMX+DART. We emphasize that the results in Fig. 21 are for a single altitude region (90 km). The comparisons would likely be quite different at higher altitudes where, for example, there is evidence of larger DW1 amplitudes during January 2010 in WACCMX+DART than in either NAVGEM-HA or JAGUAR-DAS. Further intercomparison of results among these (and possibly other) high-altitude meteorological analyses are needed to expand upon the initial results presented here. Nevertheless, the differences noted here in Figs. 18-21 indicate that the choice of high-altitude meteorological data set to constrain day-to-day meteorological variations in whole atmosphere models related to diurnal and semi-diurnal tides (either migrating or non-migrating) may impact the results, particularly in the equatorial regions. Thus, we advise users of these analyses to compare results to observations and/or other analyses to increase confidence. Further investigations where these types of differences are incorporated into constrained or "nudged" whole atmosphere model simulations as a source of uncertainty may be helpful to better quantify the impact of meteorological activity on day-to-day variations in the T/I system.

## 5. Summary and Discussion

Based on the results of this intercomparison among four analysis systems that assimilate middle atmospheric satellite observations, we find that there is overall good agreement in the latitude, altitude, and time behaviour of the zonal mean temperature and zonal winds up to approximately 50 km altitude during the December 2009 – March 2010 period. This finding is consistent with the results presented in Harvey et al. (2021), which examined 10 reanalysis data sets, but only one (MERRA-2) that extended above the stratopause and assimilated middle atmospheric temperature

observations (from MLS). Also consistent with Harvey et al. (2021), we find that significant differences among the four analyses begin to emerge above 50 km altitude at low latitudes. The present intercomparison among the NAVGEM-HA, JAGUAR-DAS, and WACCMX+DART analyses shows how large inter-analysis differences can extend above 80 km altitude. As summarized in Fig. 10, the largest zonal mean temperature differences among the analyses, ranging from 10-15 K, are found near 90 km. However, we find that the largest zonal mean zonal wind differences are found not at the highest altitudes, but near 50 km altitude at the Equator (Fig. 11). This latter result highlights the fact that these middle atmosphere analyses do not currently produce a consistent description of key climatological features such as the SAO in zonal mean zonal wind near the stratopause (Kawatani et al., 2020). A recent study by Hindley et al. (2020) highlights the importance of the SAO in modulating gravity wave momentum flux into the mesosphere and lower thermosphere. Assuming the time period evaluated here is representative of broader behavior, this disagreement in the time behavior of the zonal mean zonal winds in the tropical mesosphere and lower thermosphere (Fig. 8) among the four analyses should be remedied in order to improve confidence in the use of these analyses for studies of MLT dynamics as well as for input to whole atmosphere models to constrain lower atmospheric meteorological variability.

Intercomparison of the PW and tidal features examined here finds that the representations of the Q5DW and Q2DW in the 2009-2010 NH winter period are fairly consistent among these four analyses. Important differences emerge when comparing the latitude, altitude, and time behaviour of temperature variations related to the DW1, SW2, and DE3 tides above 80 km altitude. In particular, WACCMX+DART tidal amplitudes are consistently smaller than corresponding amplitudes in the NAVGEM-HA and JAGUAR-DAS data sets over the 2009-2010 NH winter period evaluated here. This is related to additional second order divergence damping that was included in the version of WACCMX+DART used for the present study, and has subsequently been removed, leading to increased tidal amplitudes in WACCMX+DART (Pedatella et al., 2020). As Fig. 21 shows, there can be as much as a factor of 2 difference in the temperature variance associated with equatorial DW1 among the analyses at 90 km altitude over the January-March 2010 period. Further study is needed to examine possible causes of the disagreement among the analyses, focusing both on the different types of middle atmospheric observations being assimilated (e.g., temperature profiles only vs. temperatures and constituents), the assimilation methods being used (e.g., 4DVAR vs. ensemble based, retrieval vs. radiance assimilation), and the details of the model physics (e.g., gravity wave drag, radiative heating parameterizations) being employed by each system.

It is important to note that this initial intercomparison is not meant to be the final word on the characteristics of these analyses, but rather a starting point. Given the extensive effort and computational resources involved in producing these data sets, a more thorough comparison over many years is beyond the scope of the present study. We also note that the systems producing these analyses are constantly evolving in order to improve both research and operational capabilities for specifying middle atmosphere conditions. Ultimately, more extensive intercomparisons that examine both seasonal and interannual variability of key middle atmospheric features (e.g., upward propagating waves and tides, SSWs and mesospheric coolings) over many years using the most recent version of the data available will be needed

in the future. The aim of this study is to provide some initial insight on where efforts to improve these systems could be most useful. One area for improvement highlighted in this study is in the representation of the equatorial SAO in the upper stratosphere and lower mesosphere. This effort would be facilitated in the future by ensuring that these high-altitude meteorological analysis systems routinely save fields quantifying the parameterized sub-grid scale gravity wave drag.

To further pursue improvements in these middle atmospheric meteorological systems, a follow-on validation study is planned where independent (i.e., not assimilated) satellite and ground-based middle atmosphere observations are used to evaluate each of these data sets. Some examples of independent ground-based observations for validation of middle atmospheric analyses include mesospheric horizontal wind profiles derived from meteor radars (e.g., Stober et al., 2020) and temperature profiles from lidar (e.g., Marlton et al., 2020). Some examples of independent satellite-based observations that have been used for validation include wind observations from the TIMED Doppler Interferometer (TIDI; Dhadly et al., 2019), and constituent profiles from the Solar Occultation for Ice Experiment (SOFIE; Siskind et al., 2019). A future validation study would greatly benefit from interaction with existing groups such as the Network for the Detection of Atmospheric Composition Change (NDAAC; Marlton et al., 2020) and the Atmospheric dynamics Research Infrastructure in Europe (ARISE; Blanc et al., 2017). Lastly, we would also encourage participation from other research centers producing middle atmosphere analyses in any follow-on studies motivated by the present work under the auspices of the SPARC Data Assimilation Working Group or similar organizations.

**Acknowledgements**

JPM and FS acknowledges support from NASA award 80NSSC20K0628 and from the Naval Research Laboratory Base Program. VLH acknowledges NASA grants 80NSSC18K1046 and 80NSSC19K0834. Production of NAVGEM-HA data was supported by a grant of computer time from the Department of Defense High Performance Computing Modernization Program. NAVGEM-HA analyses for the 2009-2010 winter period are available at https://map.nrl.navy.mil/map/pub/nrl/navgem/iap/. MERRA-2 analysis fields are available from the NASA Goddard Earth Sciences (GES) Data and Information Services Center (DISC), with this study's model level MERRA-2 fields at https://disc.gsfc.nasa.gov/datasets/M2I3NVASM_5.12.4/summary. LC was supported by the NASA Modeling, Analysis, and Prediction program. Resources for production of MERRA-2 were provided by the NASA High-End Computing (HEC) Program through the NASA Center for Climate Simulation (NCCS) at Goddard Space Flight Center. WACCMX+DART wind and temperature output for December 2009 to March 2010 is publicly available at https://doi.org/10.5065/d88c-y005. The data from JAGUAR-DAS is available on request.

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

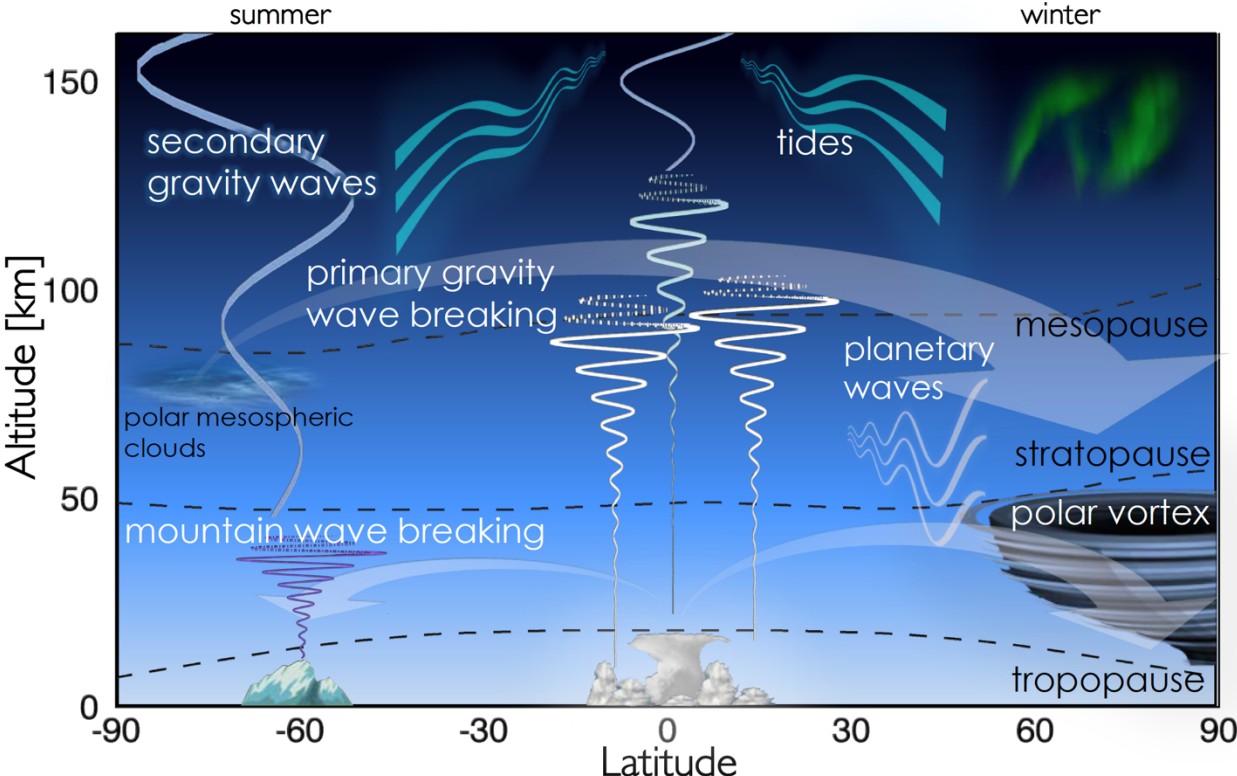

Figure 1. Sources of meteorological variability in the middle atmosphere impacting the thermosphere/ionosphere system.

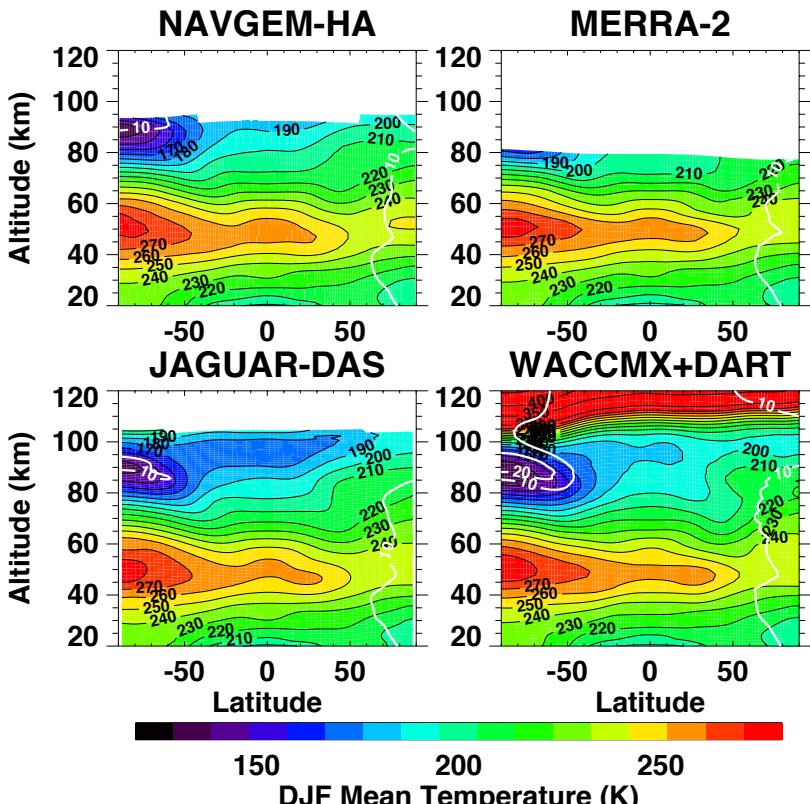

Figure 2. Latitude-altitude cross-sections of DJF 2009-2010 average zonal mean temperature in NAVGEM-HA, MERRA-2, JAGUAR-DAS, and WACCMX+DART. Thick white contours are temperature standard deviation values of 10 and 20 K.

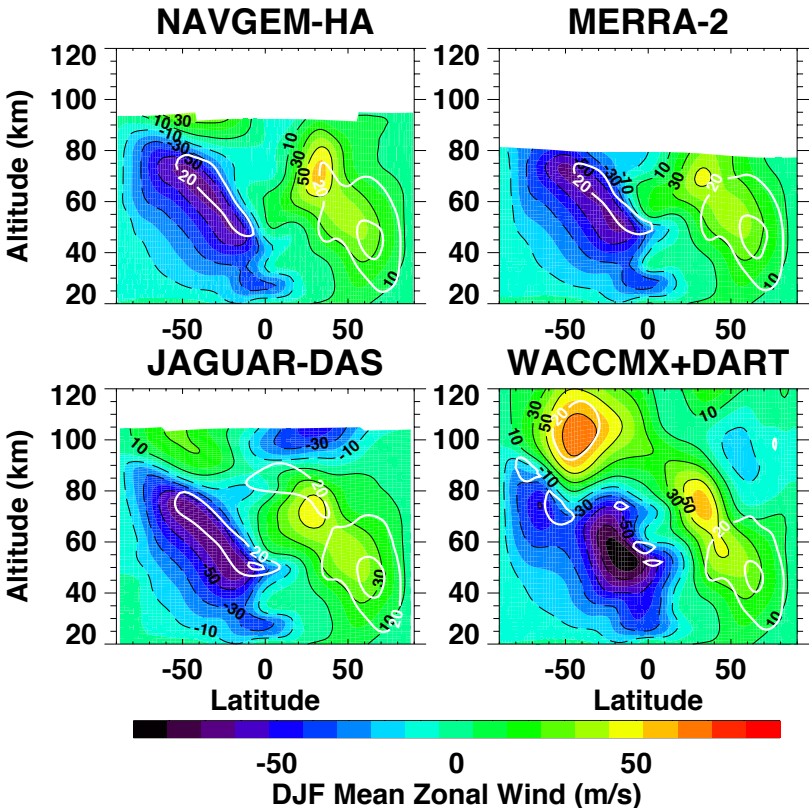

Figure 3. As in Figure 2 but for zonal wind. Dashed black contours depict easterly winds. Thick white contours are zonal wind standard deviation values of 20 and 30 m/s.


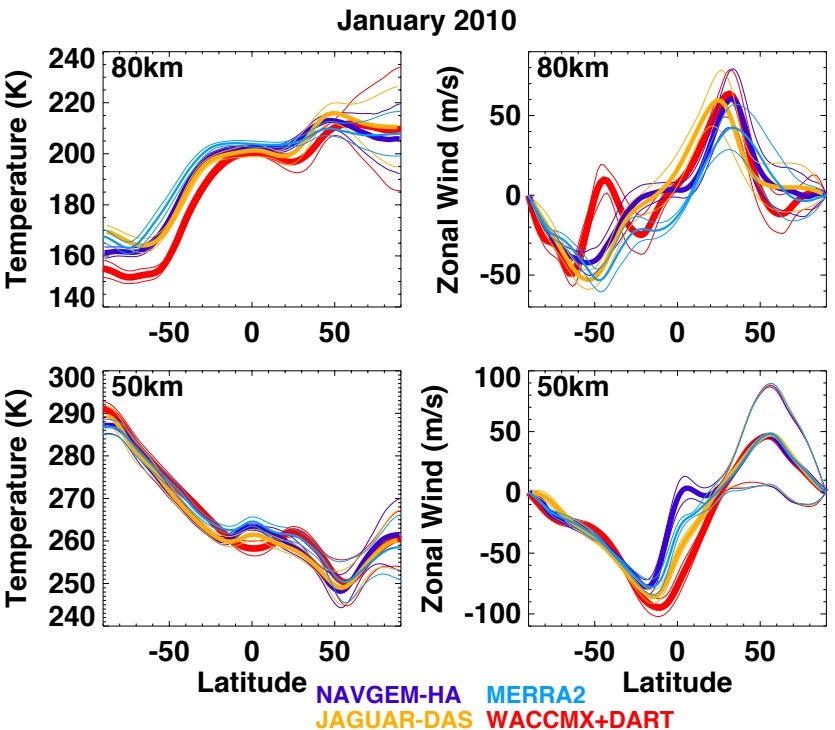

Figure 4. Latitude dependence of January 2010 average zonal mean temperature (left) and zonal wind (right) at 80 km (top) and 50 km (bottom) for NAVGEM-HA (purple), MERRA-2 (light blue), JAGUAR-DAS (gold), and WACCMX+DART (red). Thick curves indicate the monthly zonal mean values, thin curves indicate ±1 standard deviation of the daily means.

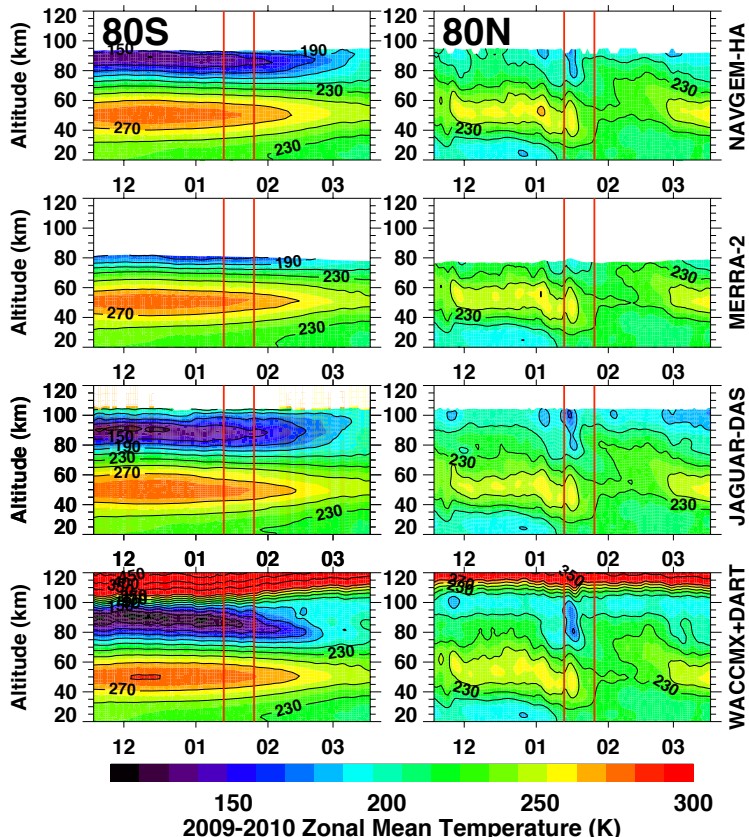

Figure 5. Altitude-time cross-sections from 1 December 2009 to 31 March 2010 of daily mean zonal mean temperature in NAVGEM-HA, MERRA-2, JAGUAR-DAS, and WACCMX+DART at 80°S (left column) and 80°N (right column). Red vertical lines in each panel denote 27 January (the onset of sustained easterly flow in the mesosphere) and 9 February (the onset of easterly flow in the stratosphere), as described in the text. Month tick labels along the x-axes are placed at the 15th of each month.


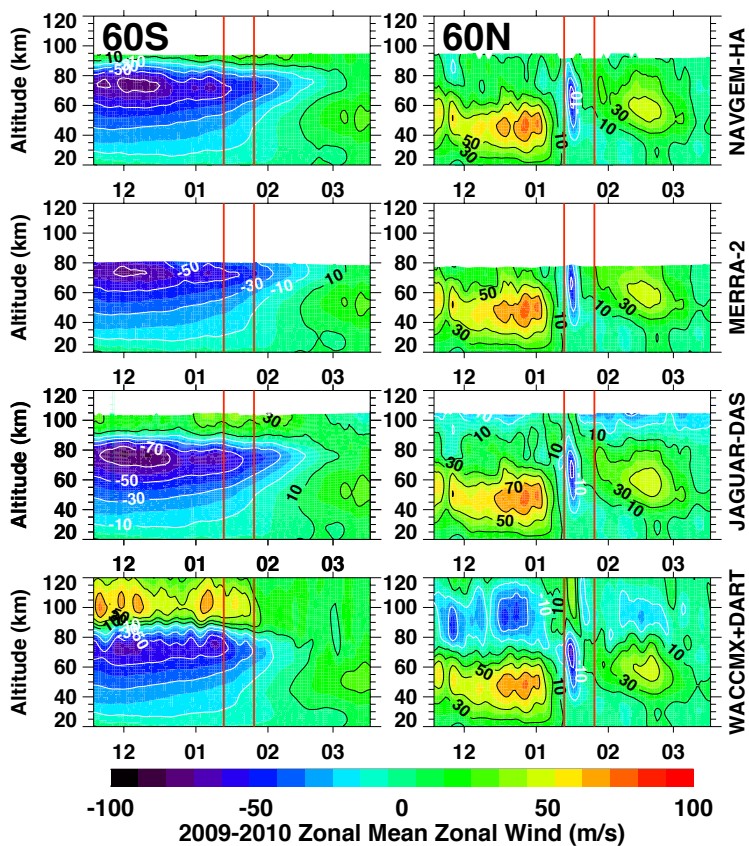

Figure 6. As in Figure 5 but for zonal wind at 60°S (left) and 60°N (right).

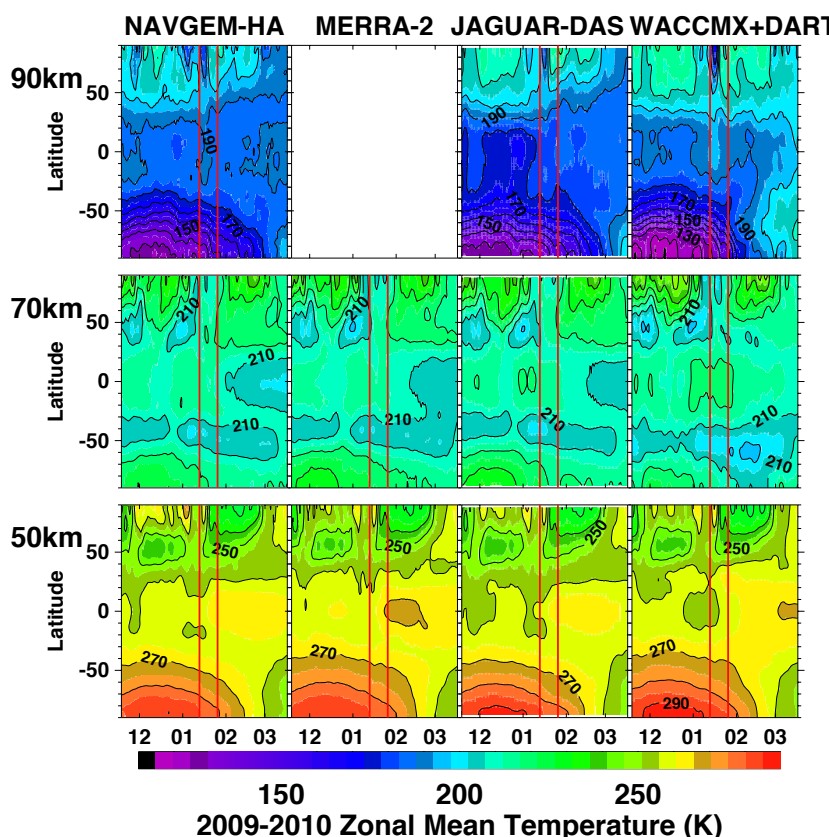

Figure 7. Latitude-time cross-sections from 1 December 2009 to 31 March 2010 of daily mean zonal mean temperature in NAVGEM-HA, MERRA-2, JAGUAR-DAS, and WACCMX+DART at 90 km (top), 70 km (middle), and 50 km (bottom). Contours are drawn every 20 K. Red vertical lines in each panel denote 27 January and 9 February, as described in the text. Month tick labels along the x-axes are placed at the 15th of each month.

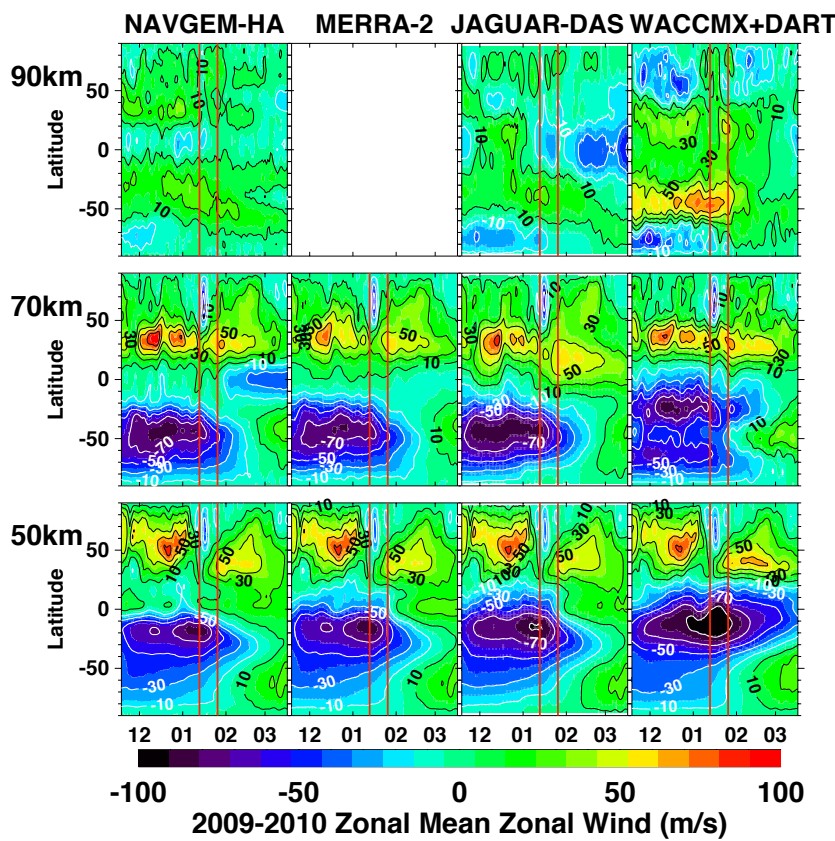

Figure 8. As in Figure 7 but for zonal wind. Contours are drawn every 20 m/s.


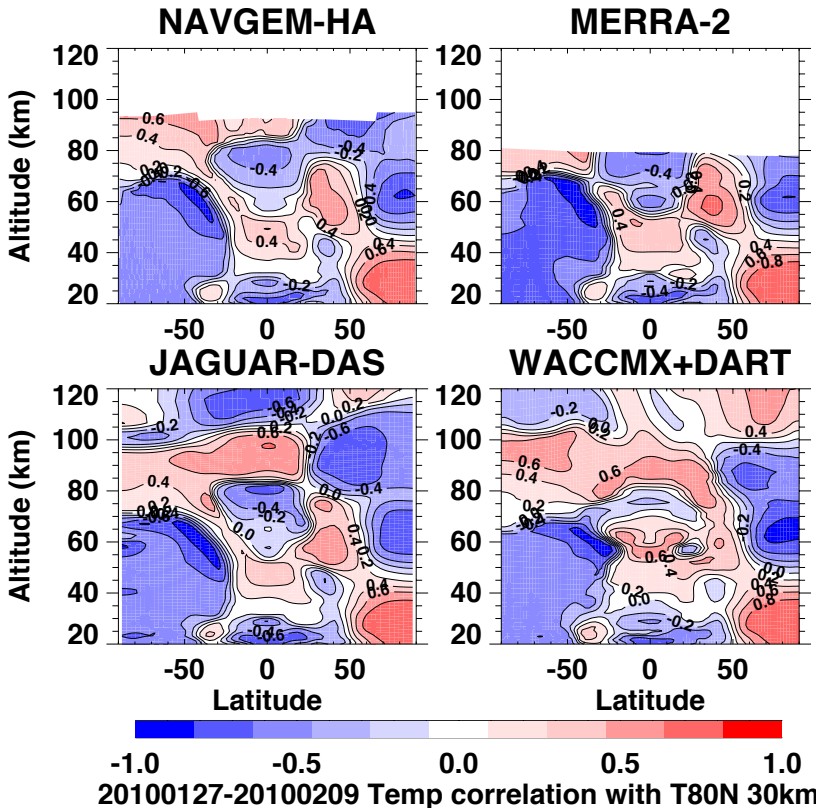

Figure 9. Latitude-altitude cross-sections of the correlation coefficient between daily zonal mean temperature at 30 km and 80º N and daily zonal mean temperature at all other latitudes and altitudes in NAVGEM-HA, MERRA-2, JAGUAR-DAS, and WACCMX+DART. The SSW disturbance time period over which the correlation coefficient is calculated is from 27 January to 9 February.

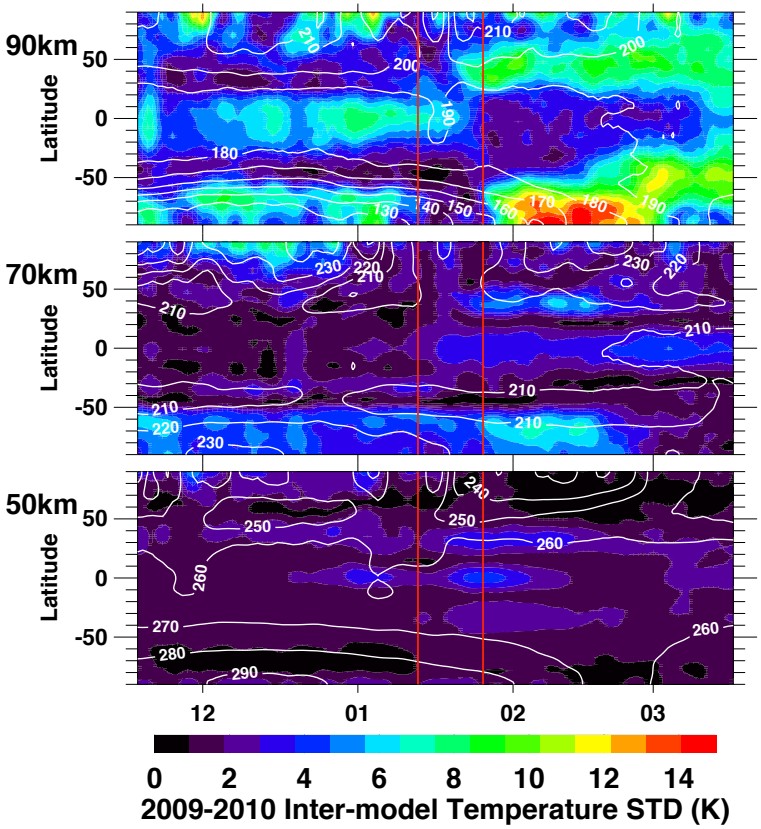

Figure 10. Latitude-time cross-sections from 1 December 2009 to 31 March 2010 of the standard deviation in daily mean zonal mean temperature among the meteorological data sets at 90 km (top), 70 km (middle), and 50 km (bottom). There is no MERRA-2 data at 90 km. For reference, white contours indicate the mean values among the data sets. Red vertical lines in each panel denote 27 January and 9 February, as described in the text. Month tick labels along the x-axes are placed at the 15th of each month.


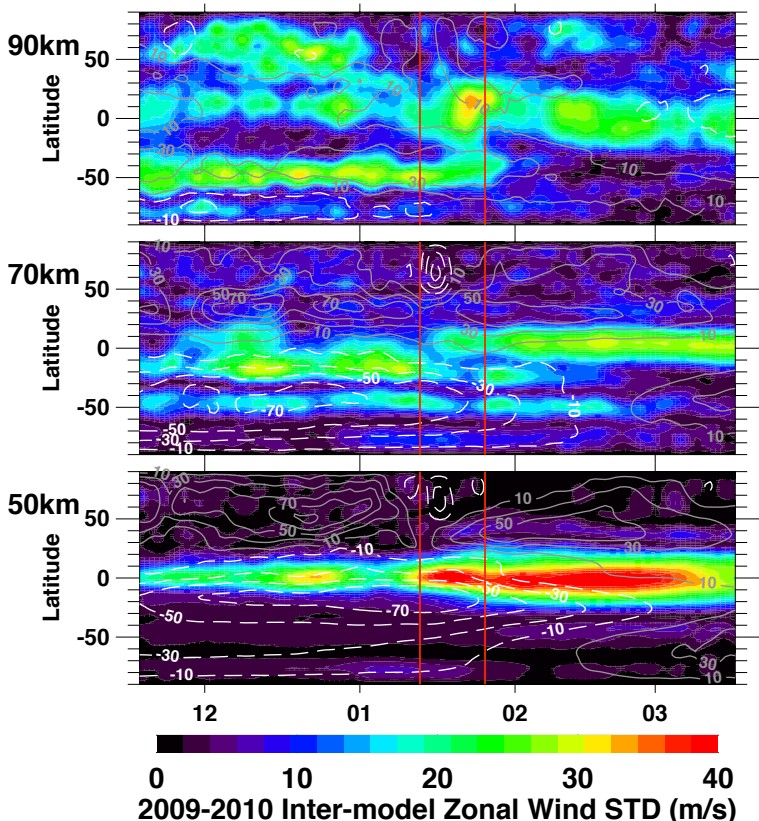

Figure 11. As in Figure 10 but for zonal wind standard deviation among the analyses. For reference, dashed white and solid gray contours indicate the mean easterly and westerly winds, respectively, among the data sets. Contours are drawn every 20 m/s.

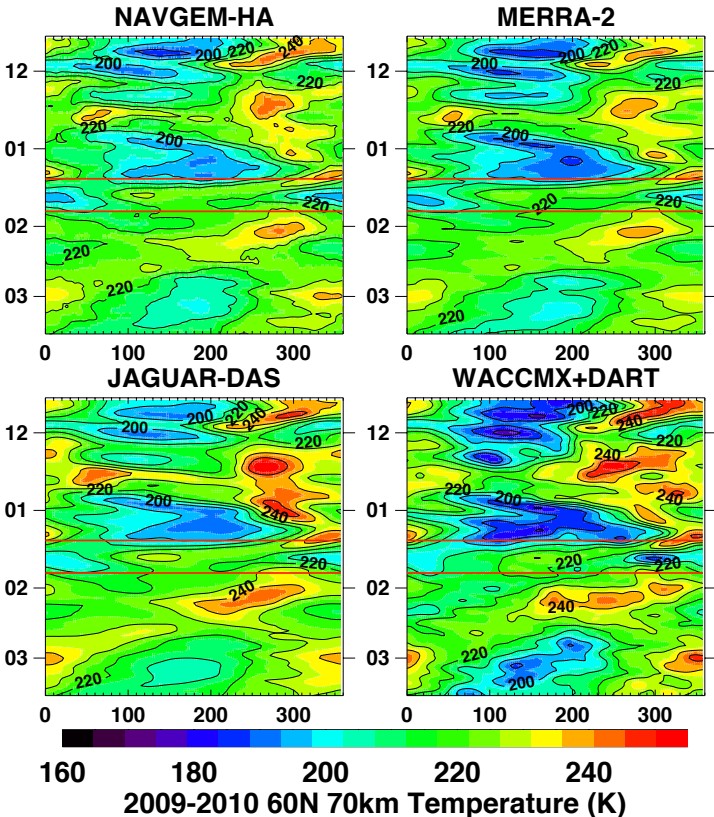

Figure 12. Longitude-time Hovmöller diagrams at 60°N and 70 km from 1 December 2009 to 31 March 2010 of daily mean temperature in NAVGEM-HA, MERRA-2, JAGUAR-DAS, and WACCMX+DART. Red horizontal lines in each panel denote 27 January and 9 February, as described in the text. Month tick labels along the y-axes are placed at the 15th of each month.

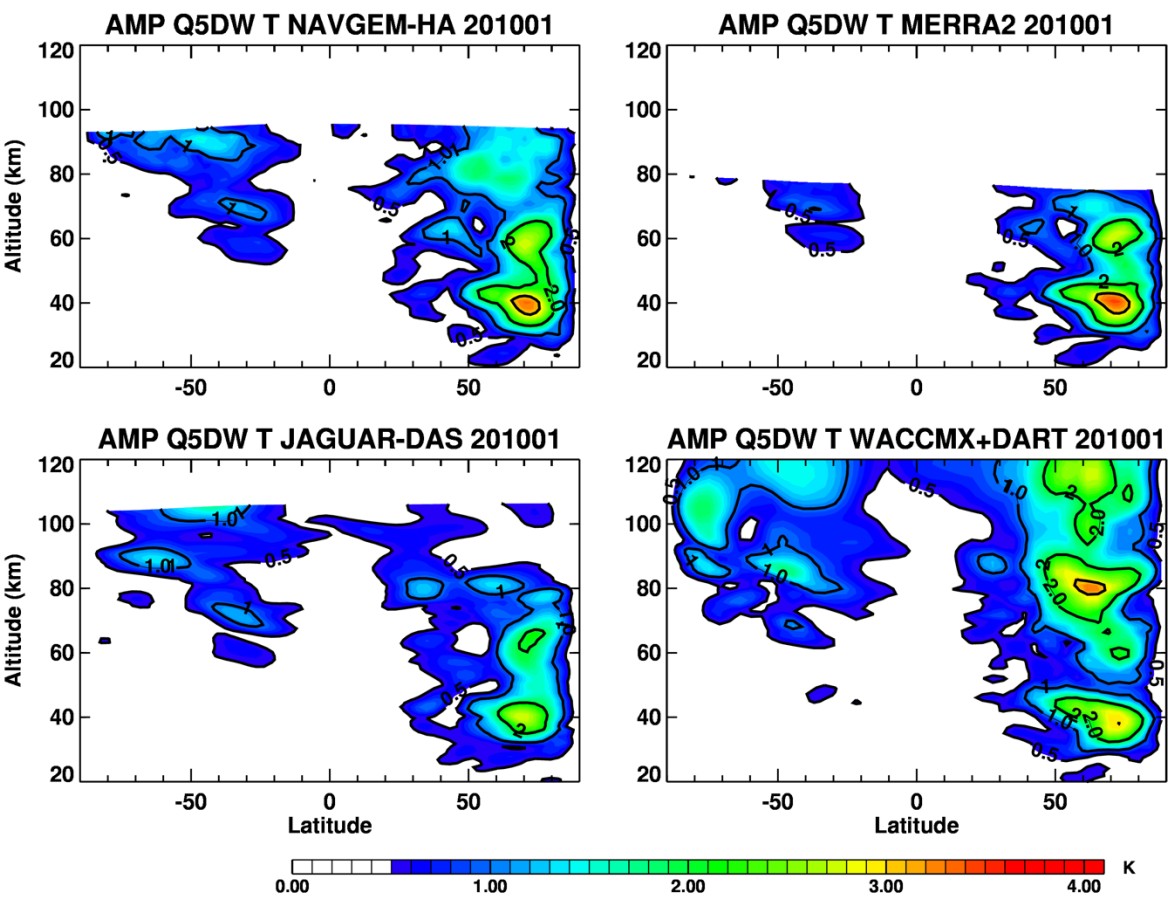

Figure 13. Monthly mean amplitude of the quasi-5 day (Q5DW) wave in temperature for January 2010 from NAVGEM-HA, MERRA-2, JAGUAR-DAS, and WACCMX+DART.


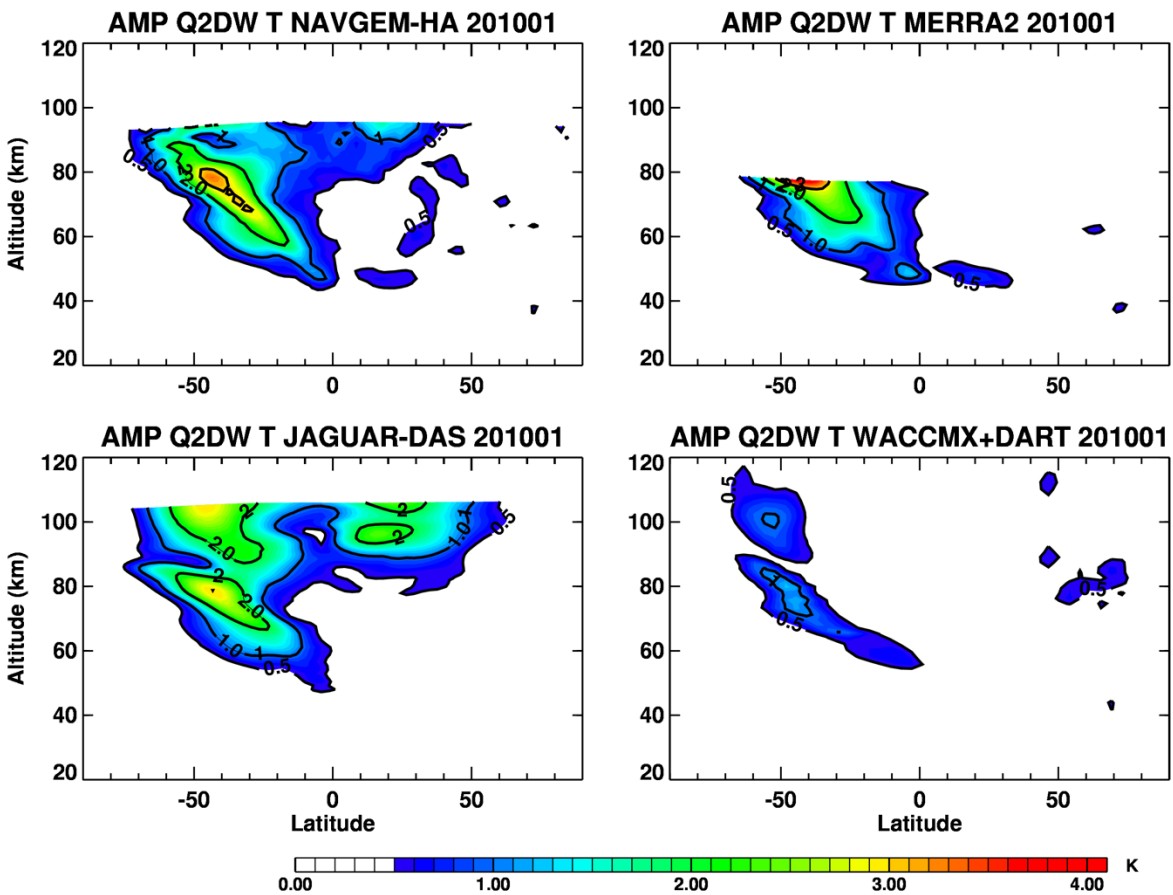

Figure 14. Monthly mean amplitude of the westward zonal wave number 3 quasi-2 day (Q2DW) wave in temperature for January 2010 from NAVGEM-HA, MERRA2, JAGUAR-DAS, and WACCMX+DART.

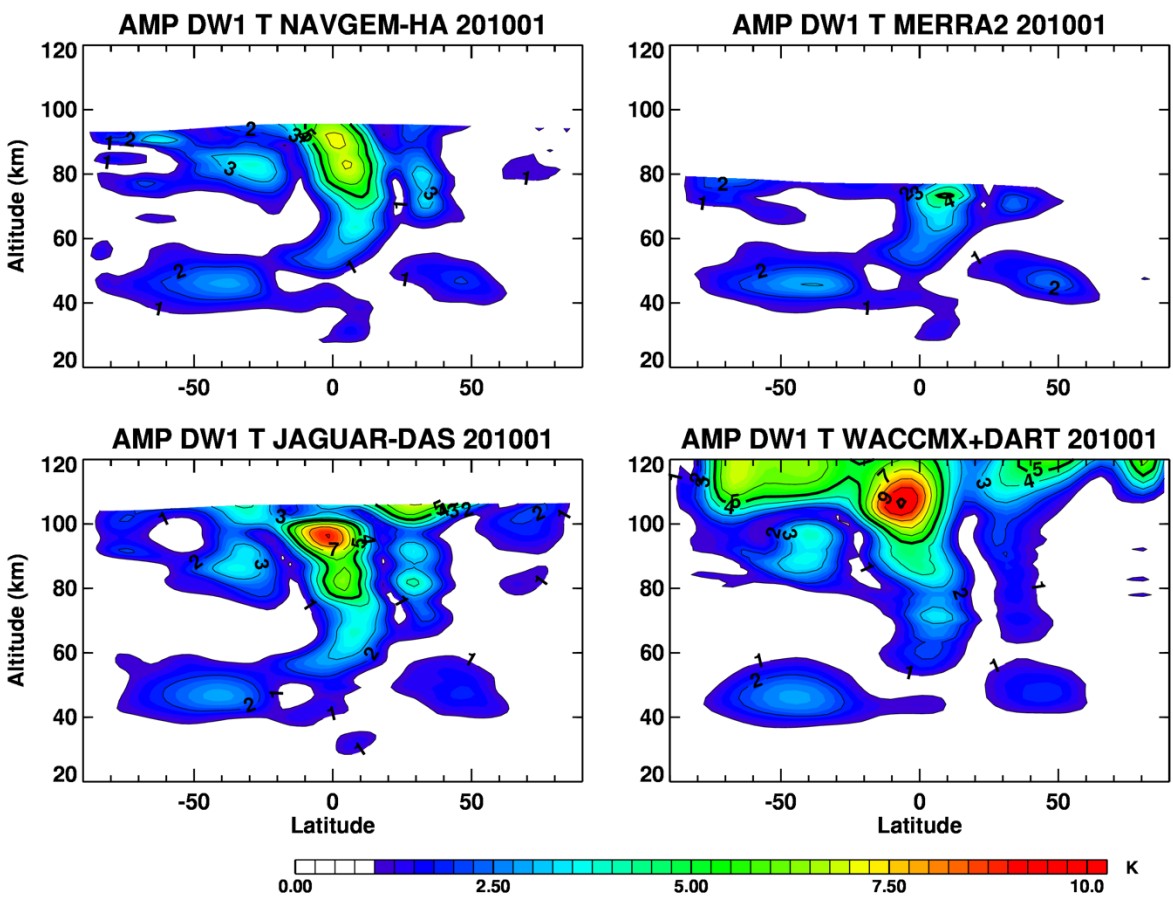

Figure 15. Monthly mean amplitude of the migrating diurnal tide (DW1) in temperature for January 2010 from NAVGEM-HA, MERRA2, JAGUAR-DAS, and WACCMX+DART.


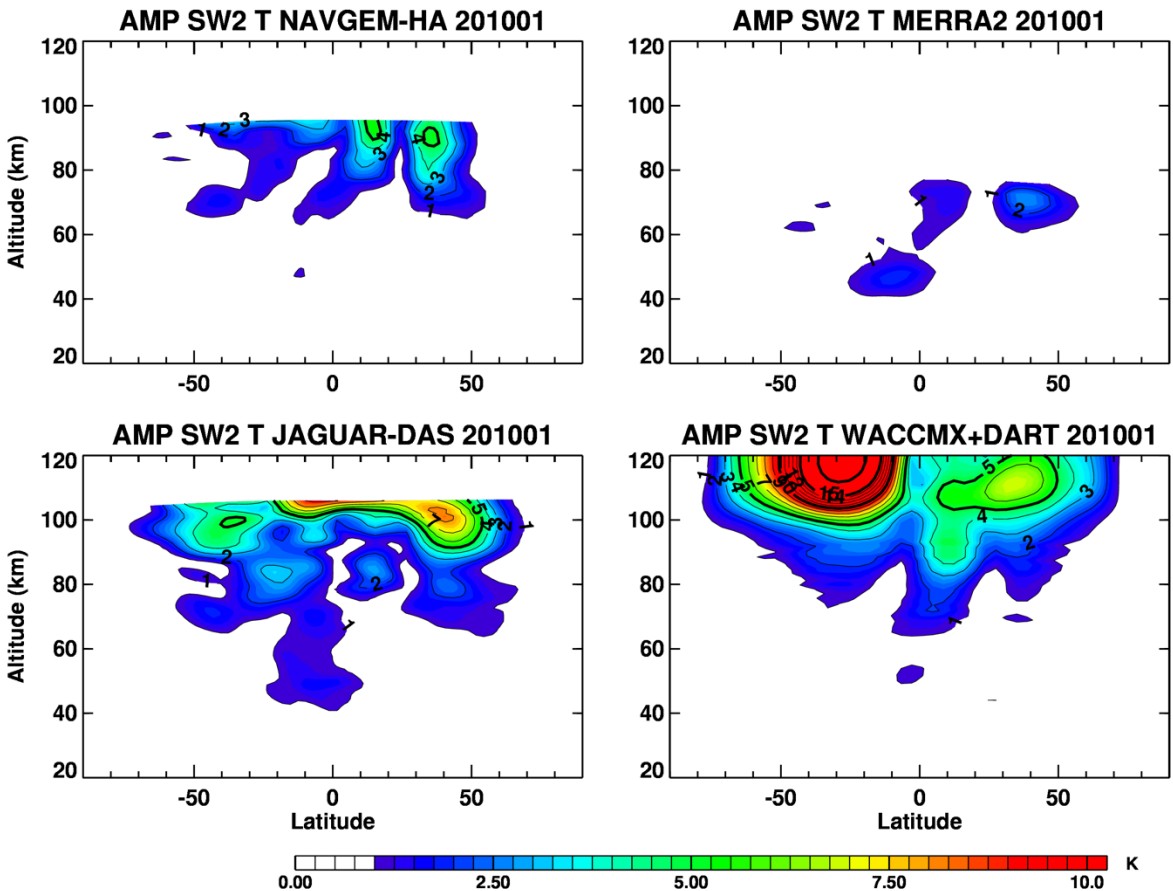

Figure 16. Monthly mean amplitude of the migrating semi-diurnal tide (SW2) in temperature for January 2010 from NAVGEM-HA, MERRA2, JAGUAR-DAS, and WACCMX+DART.

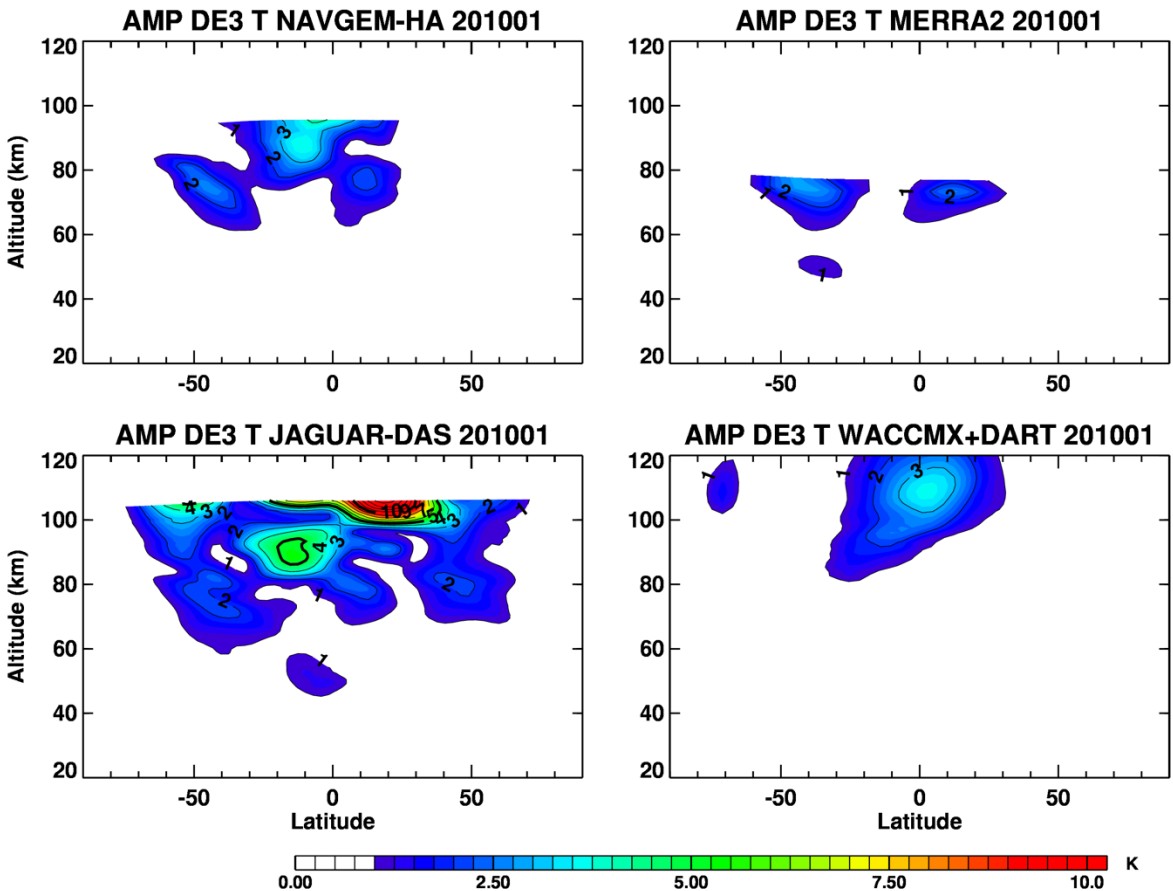

Figure 17. Monthly mean amplitude of the non-migrating wave 3 diurnal tide (DE3) in temperature for January 2010 from NAVGEM-HA, MERRA2, JAGUAR-DAS, and WACCMX+DART.

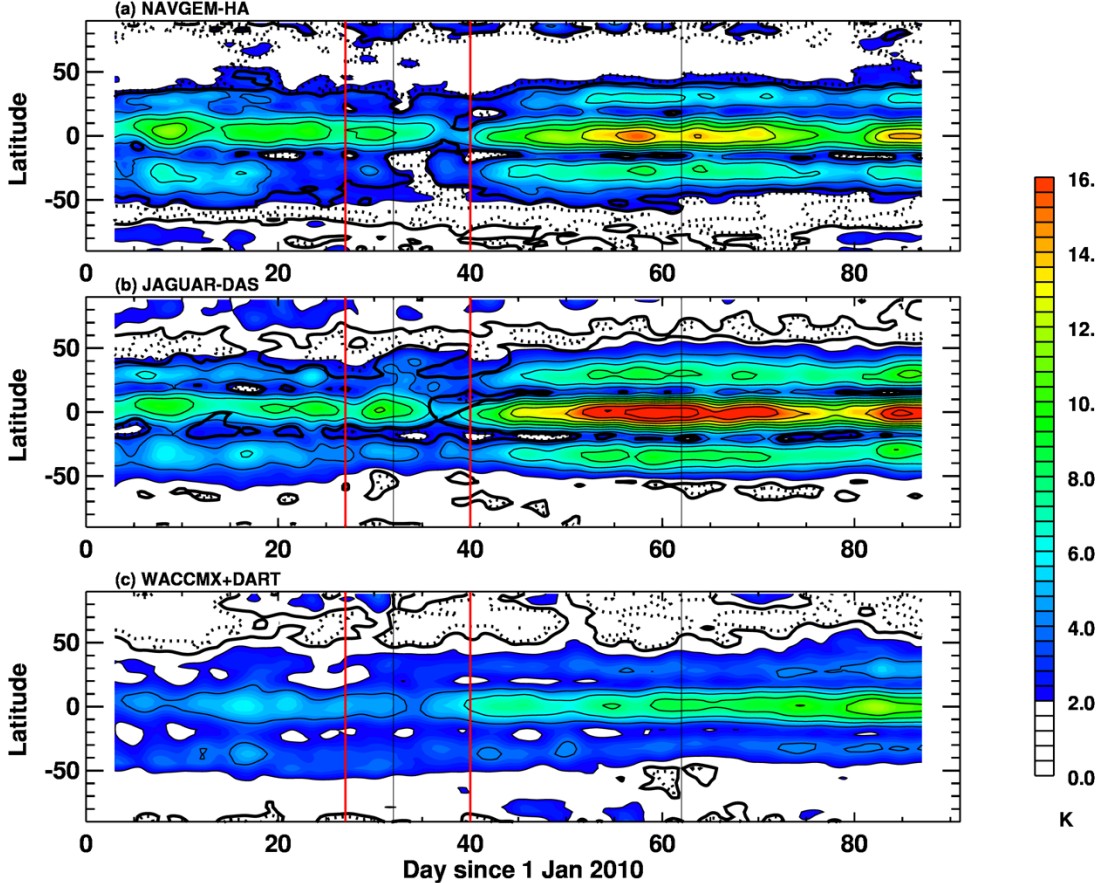

Figure 18. Latitude-time sections of the amplitude in migrating diurnal wave 1 (DW1) temperature variations at 90 km altitude from (a) NAVGEM-HA, (b) WACCMX+DART and (c) JAGUAR-DAS for Jan.-Feb.-Mar. 2010. Thin, black, solid contours are drawn every 2 K. Bold dashed and solid black contours indicate regions where results from wavelet analysis exceed 90% and 95% confidence levels. Red vertical lines in each panel denote 27 January and 9 February, as described in the text. Black vertical lines in each panel denote 1 February and 1 March.


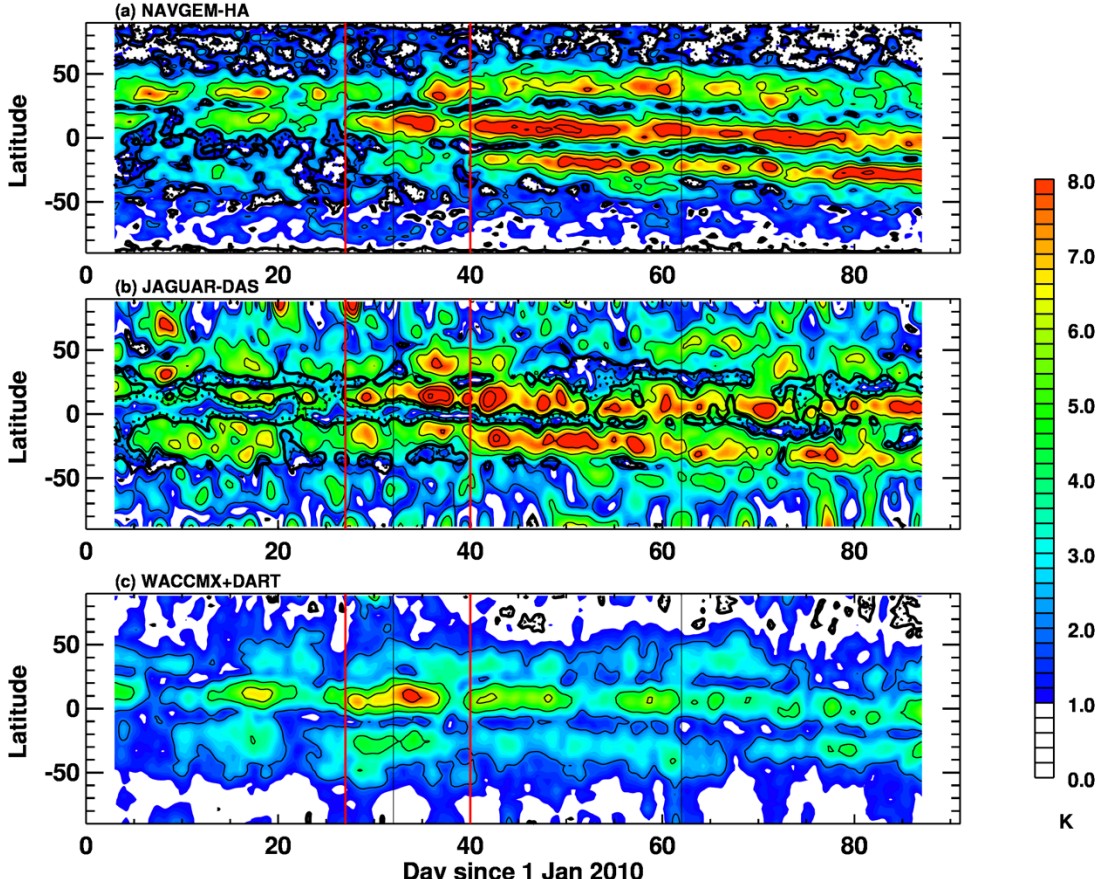

Figure 19. As in Figure 18 but for the migrating semi-diurnal wave 2 (SW2).

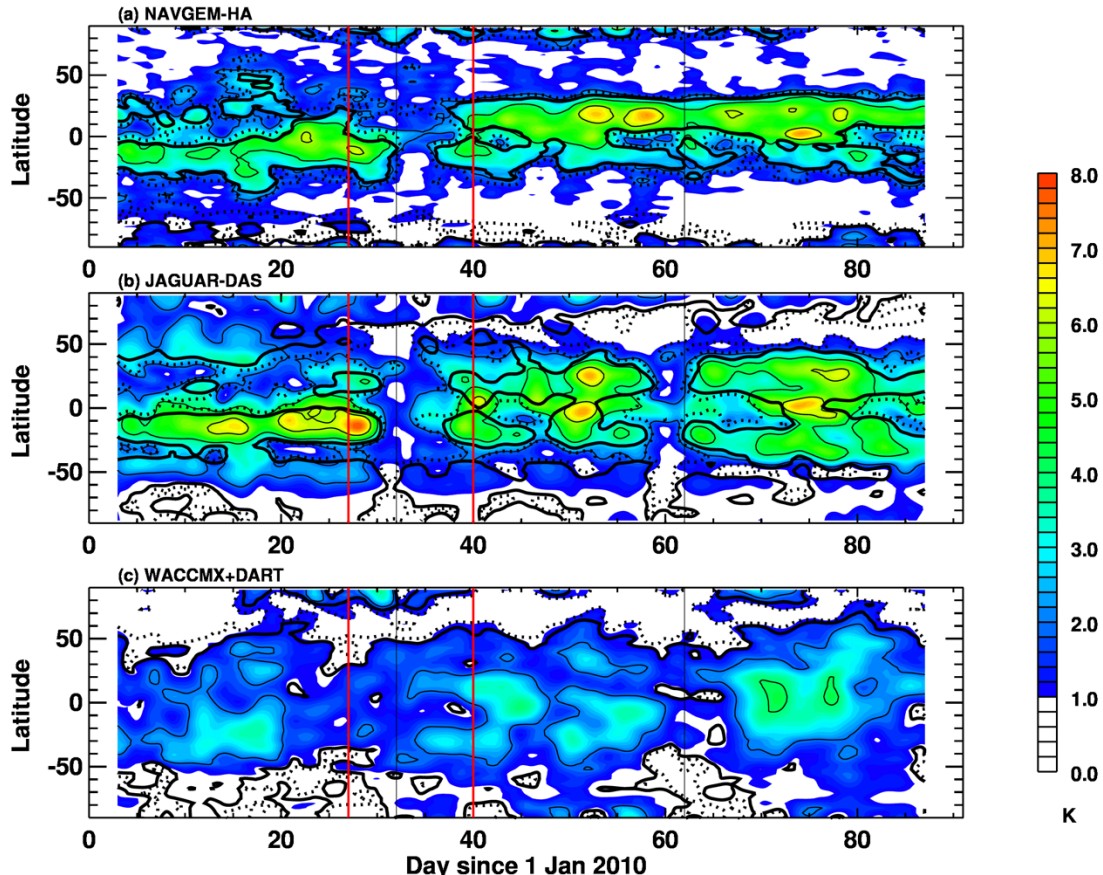

Figure 20. As in Figure 18 but for the non-migrating diurnal wave 3 (DE3).


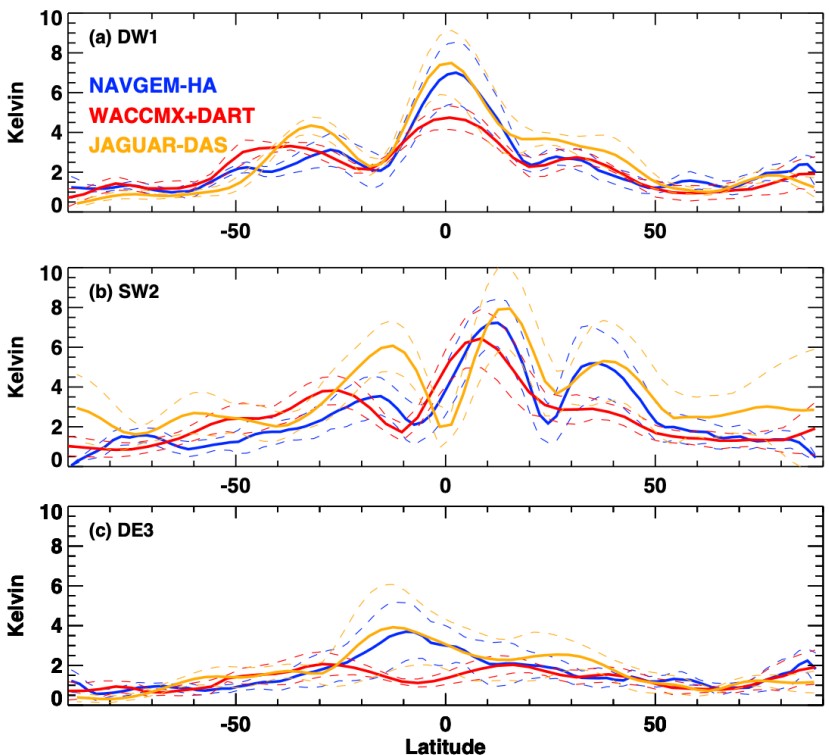

Figure 21. Latitude dependence of the mean amplitude and ±1 standard deviation values for (a) migrating diurnal wave 1 (DW1), (b) migrating semidiurnal wave 2 (SW2), and (c) non-migrating diurnal wave 3 (DE3) at 90 km altitude obtained from NAVGEM-HA, WACCMX+DART, and JAGUAR-DAS from January 27 to February 9 from Figs. 18, 19, and 20.