# Peer review of "Intercomparison of Middle Atmospheric Meteorological Analyses for the Northern Hemisphere Winter 2009-2010"

_Atmospheric Chemistry and Physics, 2021_

## Author Comment (AC1)

**Response to Referee 1**

We thank the referee for the very helpful comments and suggestions. We have revised the manuscript in response to each of the comments, as discussed in detail below. The main revisions are the addition of a table describing the basic characteristics of each system, and inclusion of confidence estimates for the S-transform results (now Figs 18-20). In response to another referee, we have also added a new figure (now #9) showing correlation patterns among temperatures referenced to 80°N latitude and 30 km altitude during the two weeks leading up to the SSW, defined as 27 Jan – 9 Feb 2010. The other referee also suggested modifying the original figure 20 to show mean amplitudes of the tides over the SSW disturbance period. See our new Fig. 21 that addresses this suggestion.

Our detailed responses to individual comments are given below in blue italics.

**L19:** "can" consistently represent ? If you would mean that they are currently consistent, it seems to be contradictory to the "large discrepancies" in L29.

We apologize for the confusion. This sentence now reads "However, the extent to which independent middle atmosphere analyses differ in their representation of wave-induced coupling to the ionosphere is unclear."

Fig. 1: The "tropopause" is at the ground.

We have revised Figure 1 to include black dashed lines that denote the tropopause, stratopause, and mesopause.

**L64**: I would suggest removing "(non-orographic)" because this could be misleading (as there exist primary orographic waves in general).

We have removed the phrase in parentheses.

L72: Please remove the comma after "predict".

All commas were removed.

L96: "in the equatorial region the lack of wind measurements ..."

We have revised the text so that it now reads "In addition, the lack of wind measurements at low latitudes..."

L100: "that" to "than"

Done.

**L117**: Maybe one of the two "constrained" might be removable.

Done.

**L154-155**: "which previous ... system." : This seems to need to be rephrased.

We have revised the sentence to clarify the meaning, i.e., that previous studies cited in Section 1 have shown the importance of PWs and tides for day-to-day variability in the thermosphere and ionosphere.

**L159**: "horizontal and vertical" or "zonal and meridional" ? (By the way, has there been a place where v or w is used for this intercomparison ?)

Thank you for catching this typo. This sentence now reads "For this intercomparison, we examine global gridded data sets of temperature, zonal wind and geopotential height..."

L183: "produces global synoptic gridded atmospheric data sets" : repetitive (L159)

We have removed "global synoptic gridded" to avoid unnecessary repetition.

**Section 2.1-2.4**: Horizontal and vertical resolutions of the four models : Would these be easier to read from a table rather than from the text in each section ?

|                 |                                                     | Horizontal Grid,        | Vertical Range              | Reference(s) for                     |
|-----------------|-----------------------------------------------------|-------------------------|-----------------------------|--------------------------------------|
| Analysis System | Reference(s)                                        | Vertical Grid, and      |                             | Gravity Wave Drag                    |
|                 |                                                     | Output Frequency        |                             | Parameterizations                    |
| JAGUAR-DAS      | Koshin et al. (2020);
Koshin et al (in review)   | 2.8125° lat/lon,        | Surface to                  | ORO: McFarlane (1987)                |
|                 |                                                     | ∆z≈1 km,                | 1 x 10⁻ 6 hPa    | NON: Hines (1997); Watanabe          |
|                 |                                                     | ∆t=6 hours              | (~150 km)                   | (2008)                               |
| MERRA-2         | Bosilovich et al. (2015);                           | 0.625° lon by 0.5° lat, | Surface to                  | ORO: McFarlane (1987)                |
|                 | Gelaro et al. (2017);                               | ∆z≈2-5 km,              | 0.01hPa                     | NON: Garcia & Boville (1994);        |
|                 | Molod et al. (2015)                                 | ∆t=3 hours              | (~75 km)                    | Molod et al. (2015)                  |
| NAVGEM-HA       | McCormack et al. (2017);
Eckermann et al. (2018) | 1º lat/lon,             | Surface to                  | OBO: Webster et al. (2002)           |
|                 |                                                     | ∆z≈2-4 km,              | 6 x 10 -5 hPa    | NON: Eckermann (2011)                |
|                 |                                                     | ∆t=3 hours              | (~115 km)                   |                                      |
| WACCMX+DART     | Liu et al. (2018);
Pedatella et al. (2018)       | 2º lat/lon,             | Surface to                  | ORO: McFarlane (1987)                |
|                 |                                                     | ∆z≈1-5 km,              | 4.1 x 10 -10 hPa | NON: Beres et al. (2005); Richter et |
|                 |                                                     | $\Delta t=1$ hour       | (~500-700 km)               | al. (2010); Garcia et al. (2017)     |

We have added the following table for added clarity.

**Table 1.** List of analysis datasets used in this paper, overall references describing each system, the horizontal, vertical, and temporal characteristics of the analysis output, the model top, and references for gravity wave specifications. In the 5th column, ORO refers to the parametrization for orographic gravity waves while NON refers to that of non-orographic gravity waves.

**L226**: "for the surface" to "from the surface"

Done.

L227-228: Maybe the second "system" (L228) should be removed.

Done. This sentence now reads, "The JAGUAR model generates short-term forecasts that are used as background fields for the data assimilation system (JAGUAR-DAS), which employs a fourdimensional local ensemble transform Kalman filter (4D-LETKF) developed by Miyoshi and Yamane (2007)."

L261: "additional" : to what ? (Or just remove this word.)

We have removed "Additional...". The second order divergence damping is increased in the thermosphere and is the only divergence damping applied, so it is not additional.

**L299**: " $\gamma_45$  is ... latitude" on the surface ?

*Yes, this refers to the surface value, which is now stated explicitly.*

Fig. 2 caption: I could not find the thin horizontal line in the figure.

We have deleted this sentence. The thin lines referred to earlier versions of these figures, which are not used in this manuscript.

**L330-331**: "Between 50 km and 80 km" : the summer/winter polar mesopause explained in this sentence exists outside of this range of altitude.

This has been revised to read "Near 80 km altitude,..."

L355: "0 deg S" to "0 deg" (or "0")

Changed OS to "Equator"

**L353-366**: Regarding the mean wind (and temperature) bias in WACCMX+DART : Could you please provide some potential reasons or relevant references for this bias, if possible ?

In the revised text we now note that biases in the WACCMX background model were found in the study Marsh et al. (2013) and are most likely due to errors in the GW parameterizations. This work shows that these known wind biases are not fully corrected by the assimilation of stratospheric and mesospheric temperature observations, since this depends on how well the temperature observations can constrain the winds.

L382: "... state." : A reference would be helpful.

We have included a reference to the 2003 Reviews of Geophysics article by Fritts and Alexander.

**L384-385 (also in L464 and L495)**: "suggesting that differences in the treatment of gravity wave drag may be a primary factor explaining the large differences among the analyses above 80 km" : Provided the existence of the difference in parameterized GW processes between models

(which I fully agree), what would be the role of the differences in the data assimilation methods ? Would the larger difference between WACCMX+DART and the others than the difference among the latter three (shown in Figs. 2 and 3) mean that the difference in GW parameterization is also larger for WACCMX+DART from the others ?

We have revised the sentence to state that differences in the treatment of GWD may be an "important factor", rather than a "primary" factor as originally stated. Since it is well-established that GWD plays a key role in determining the distribution of winds and temperatures in this region, this is a reasonable statement. We also cite Pedatella et al. (2014a) who showed that gravity wave drag differences among models is related to differences in the background winds. Unfortunately, we are not currently able to say more about the role of the differences with regards to the data assimilation methods, since we lack key output from the systems regarding wind increments and resolved vs. parameterized GW effects within the model component. These quantities are not routinely archived due to limited computational/storage resources. Therefore, as now stated in the paper, any detailed attribution is unfortunately beyond the scope of this initial intercomparison.

L420: "eastward" : Would it be "westward" ?

Yes, we have changed to "westward".

**L500**: I would suggest changing "Difference among ... in the analyses" to "The spread of ... among the analyses".

We have revised the sentence to now begin "To examine the range in zonal mean temperatures and zonal winds,...".

L516: "addition"

Corrected.

L521: "in daily-mean temperature at ..."

**Corrected.**

**L548**: "in each hemisphere during summer" : It was not clear to me. Is it mean the northern summer ? Or, the solstices ?

We have revised this sentence to read "in the summer hemisphere" to clarify that we are referring to the respective summer solstice periods in each hemisphere.

**L554**: "4-6 days" ? (or 0.24 cpd ?)

Yes, we have revised this to the correct value of 0.24 cpd.

L555: "high-altitude" to "high-latitude" ?

Yes, we have revised this to read "high-latitude".

L599: "seasonal" to "altitude" ?

Yes, thank you! "seasonal" was changed to "altitude".

**L601-609 and L639-641**: In these lines, the day-to-day variations of tides associated with SSWs in previous studies are being discussed. I would suggest moving this content to L653 where I see it is more relevant (with Figs. 17-19), as the current paragraphs are explaining the monthly mean amplitudes of tides.)

The discussion in original lines 601-609 has been moved to later in this section where day-to-day tidal variability is discussed. We have retained the discussion on original lines 639-641 regarding DE3 as we feel it provides necessary motivation to compare DE3.

L611-612: "latitude and ... January 2010" : repetitive (L599-600)

This has been reworded to avoid repetition.

**L620-622**: The peak amplitude of the tide seems to appear slightly (5-10 km) below the top of each model. Would it be related to some damping mechanisms near the model top ?

This is potentially an issue for the lower top MERRA2 and NAVGEM-HA systems, but not for WACCMX+DART and JAGUAR-DAS whose model tops are near 500 km and 150 km, respectively. Theoretically, the upward propagating component of DW1 should peak around ~80-90 km, and there is another peak in DW1 generated in-situ in the thermosphere, which we see from 110-120 km in WACCMX+DART. The top of the WACCMX+DART system is well above these altitudes, so we do not expect damping at the model top to affect this. Likewise, the top of the JAGUAR-DAS system is well above the altitudes where DW1 peaks in theory, so effects of damping near model top are negligible.

We have added the following text to respond to this remark, "The range of altitudes for maximum DW1 amplitudes seen in these four analyses agrees with SABER observations (Zhang et al., 2006). For MERRA-2 and possibly NAVGEM-HA, the analysis system upper boundaries are low enough that artificially damping of DW1 may occur. In JAGUAR-DAS, DW1 is dissipated above ~100 km due to the model diffusion exponentially increasing with height to mimic molecular diffusion. The differences in DW1 structure at/above 100 km between JAGUAR-DAS and WACCMX+DART are likely due to the large differences in background zonal mean zonal wind (Fig. 3)."

L644: But at quite different latitudes : 35 deg S in MERRA2 and 5 deg S in JAGUAR-DAS

We thank the referee for noting this difference. We have revised the text to indicate the different latitudes where DE3 peaks near 50 km in MERRA2 (~35°S) and JAGUAR-DAS (~5°S).

**Figs. 17-19**: 1) I would suggest exchanging (b) and (c) in order to be in the consistent order with the previous figures. 2) If the vertical lines indicate the boundaries of the months, please provide this information in the caption. If they are not, please include "(Day XX)" (the x-axis value) after "early February" in L671 and L676.

Due to the addition of a figure at the suggestion of referee 2, these are now Figs. 18-20. We have reordered the figures, described in the caption that the black vertical lines denote the boundaries of the months, and we added vertical red lines to indicate the SSW warming period from 27 Jan – 9 Feb in response to Referee 2's request on this and other figures showing time variations. We have also added confidence estimates as dotted and solid black contours in response to the comment about L679 (see below).

L656-659: Could it be removed but with just referring Section 2?

Yes, we have largely re-written this paragraph.

L660-661: This might be removable. : repetitive (L654-655)

**Deleted.**

**L679**: Would it be possible to check the statistical confidence in the wavelet spectrum ? (like in Torrence and Compo, 1998, BAMS)

We have added 90% and 95% confidence estimates to new Figs. 18-20 (pasted below) based on the method outlined in Torrance and Compo (1998), using an AR1 background spectrum as described in Sassi (2012). A description of the method for obtaining these confidence estimates has been added to Section 2. Confidence estimates are generated separately for each latitude and frequency, using a background spectrum whose variance corresponds to the input temperature time series at each latitude. These results are plotted on top of the temperature amplitudes in Figs 18-20 to indicate when and where the results exceed the 90% (dashed black ) and 95% (solid black contour) limits. We find that the majority of DW1 and SW2 results fall within the 95% confidence estimates. Some of the DE3 results fall below the 90% confidence estimate in the JAGUAR-DAS results, likely a result of the much coarser time resolution (6 hourly output) compared to WACCMX\_DART and NAVGEM-HA.

---

## Author Comment (AC2)

**Response to Referee 1**

We thank the referee for the very helpful comments and suggestions. We have revised the manuscript in response to each of the comments, as discussed in detail below. The main revisions are the addition of a table describing the basic characteristics of each system, and inclusion of confidence estimates for the S-transform results (now Figs 18-20). In response to another referee, we have also added a new figure (now #9) showing correlation patterns among temperatures referenced to 80°N latitude and 30 km altitude during the two weeks leading up to the SSW, defined as 27 Jan – 9 Feb 2010. The other referee also suggested modifying the original figure 20 to show mean amplitudes of the tides over the SSW disturbance period. See our new Fig. 21 that addresses this suggestion.

Our detailed responses to individual comments are given below in blue italics.

**L19:** "can" consistently represent ? If you would mean that they are currently consistent, it seems to be contradictory to the "large discrepancies" in L29.

*We apologize for the confusion. This sentence now reads "However, the extent to which independent middle atmosphere analyses differ in their representation of wave-induced coupling to the ionosphere is unclear."*

**Fig. 1**: The "tropopause" is at the ground.

*We have revised Figure 1 to include black dashed lines that denote the tropopause, stratopause, and mesopause.*

**L64**: I would suggest removing "(non-orographic)" because this could be misleading (as there exist primary orographic waves in general).

*We have removed the phrase in parentheses.*

**L72**: Please remove the comma after "predict".

*All commas were removed.*

**L96**: "in the equatorial region the lack of wind measurements ..."

*We have revised the text so that it now reads "In addition, the lack of wind measurements at low latitudes…"*

**L100**: "that" to "than"

*Done.*

**L117**: Maybe one of the two "constrained" might be removable.

*Done.*

**L154-155**: "which previous … system." : This seems to need to be rephrased.

*We have revised the sentence to clarify the meaning, i.e., that previous studies cited in Section 1 have shown the importance of PWs and tides for day-to-day variability in the thermosphere and ionosphere.*

**L159**: "horizontal and vertical" or "zonal and meridional" ? (By the way, has there been a place where v or w is used for this intercomparison ?)

*Thank you for catching this typo. This sentence now reads "For this intercomparison, we examine global gridded data sets of temperature, zonal wind and geopotential height…"*

**L183**: "produces global synoptic gridded atmospheric data sets" : repetitive (L159)

*We have removed "global synoptic gridded" to avoid unnecessary repetition.*

**Section 2.1-2.4**: Horizontal and vertical resolutions of the four models : Would these be easier to read from a table rather than from the text in each section ?

*We have added the following table for added clarity.*

| Analysis System | Reference(s) | Horizontal Grid, Vertical Grid, and Output Frequency | Vertical Range | Reference(s) for Gravity Wave Drag Parameterizations |
|---|---|---|---|---|
| JAGUAR-DAS | Koshin et al. (2020); Koshin et al (in review) | 2.8125° lat/lon, $\Delta z \approx 1$ km, $\Delta t = 6$ hours | Surface to $1 \times 10^{-6}$ hPa (~150 km) | ORO: McFarlane (1987) NON: Hines (1997); Watanabe (2008) |
| MERRA-2 | Bosilovich et al. (2015); Gelaro et al. (2017); Molod et al. (2015) | 0.625° lon by 0.5° lat, $\Delta z \approx 2\text{-}5$ km, $\Delta t = 3$ hours | Surface to 0.01 hPa (~75 km) | ORO: McFarlane (1987) NON: Garcia & Boville (1994); Molod et al. (2015) |
| NAVGEM-HA | McCormack et al. (2017); Eckermann et al. (2018) | 1° lat/lon, $\Delta z \approx 2\text{-}4$ km, $\Delta t = 3$ hours | Surface to $6 \times 10^{-5}$ hPa (~115 km) | ORO: Webster et al. (2003) NON: Eckermann (2011) |
| WACCMX+DART | Liu et al. (2018); Pedatella et al. (2018) | 2° lat/lon, $\Delta z \approx 1\text{-}5$ km, $\Delta t = 1$ hour | Surface to $4.1 \times 10^{-10}$ hPa (~500-700 km) | ORO: McFarlane (1987) NON: Beres et al. (2005); Richter et al. (2010); Garcia et al. (2017) |

**Table 1.** List of analysis datasets used in this paper, overall references describing each system, the horizontal, vertical, and temporal characteristics of the analysis output, the model top, and references for gravity wave specifications. In the 5[th] column, ORO refers to the parametrization for orographic gravity waves while NON refers to that of non-orographic gravity waves.

**L226**: "for the surface" to "from the surface"

*Done.*

**L227-228**: Maybe the second "system" (L228) should be removed.

*Done. This sentence now reads, "The JAGUAR model generates short-term forecasts that are used as background fields for the data assimilation system (JAGUAR-DAS), which employs a four-dimensional local ensemble transform Kalman filter (4D-LETKF) developed by Miyoshi and Yamane (2007)."*

**L261**: "additional" : to what ? (Or just remove this word.)

*We have removed "Additional…". The second order divergence damping is increased in the thermosphere and is the only divergence damping applied, so it is not additional.*

**L299**: "γ_45 is ... latitude" on the surface ?

*Yes, this refers to the surface value, which is now stated explicitly.*

**Fig. 2 caption**: I could not find the thin horizontal line in the figure.

*We have deleted this sentence. The thin lines referred to earlier versions of these figures, which are not used in this manuscript.*

**L330-331**: "Between 50 km and 80 km" : the summer/winter polar mesopause explained in this sentence exists outside of this range of altitude.

*This has been revised to read "Near 80 km altitude,…"*

**L355**: "0 deg S" to "0 deg" (or "0")

*Changed 0S to "Equator"*

**L353-366**: Regarding the mean wind (and temperature) bias in WACCMX+DART : Could you please provide some potential reasons or relevant references for this bias, if possible ?

*In the revised text we now note that biases in the WACCMX background model were found in the study Marsh et al. (2013) and are most likely due to errors in the GW parameterizations. This work shows that these known wind biases are not fully corrected by the assimilation of stratospheric and mesospheric temperature observations, since this depends on how well the temperature observations can constrain the winds.*

**L382**: "... state." : A reference would be helpful.

*We have included a reference to the 2003 Reviews of Geophysics article by Fritts and Alexander.*

**L384-385 (also in L464 and L495)**: "suggesting that differences in the treatment of gravity wave drag may be a primary factor explaining the large differences among the analyses above 80 km" : Provided the existence of the difference in parameterized GW processes between models

(which I fully agree), what would be the role of the differences in the data assimilation methods ? Would the larger difference between WACCMX+DART and the others than the difference among the latter three (shown in Figs. 2 and 3) mean that the difference in GW parameterization is also larger for WACCMX+DART from the others ?

*We have revised the sentence to state that differences in the treatment of GWD may be an "important factor", rather than a "primary" factor as originally stated. Since it is well-established that GWD plays a key role in determining the distribution of winds and temperatures in this region, this is a reasonable statement. We also cite Pedatella et al. (2014a) who showed that gravity wave drag differences among models is related to differences in the background winds. Unfortunately, we are not currently able to say more about the role of the differences with regards to the data assimilation methods, since we lack key output from the systems regarding wind increments and resolved vs. parameterized GW effects within the model component. These quantities are not routinely archived due to limited computational/storage resources. Therefore, as now stated in the paper, any detailed attribution is unfortunately beyond the scope of this initial intercomparison.*

**L420**: "eastward" : Would it be "westward" ?

*Yes, we have changed to "westward".*

**L500**: I would suggest changing "Difference among ... in the analyses" to "The spread of ... among the analyses".

*We have revised the sentence to now begin "To examine the range in zonal mean temperatures and zonal winds,..".*

**L516**: "addition"

*Corrected.*

**L521**: "in daily-mean temperature at ..."

*Corrected.*

**L548**: "in each hemisphere during summer" : It was not clear to me. Is it mean the northern summer ? Or, the solstices ?

*We have revised this sentence to read "in the summer hemisphere" to clarify that we are referring to the respective summer solstice periods in each hemisphere.*

**L554**: "4-6 days" ? (or 0.24 cpd ?)

*Yes, we have revised this to the correct value of 0.24 cpd.*

**L555**: "high-altitude" to "high-latitude" ?

*Yes, we have revised this to read "high-latitude".*

**L599**: "seasonal" to "altitude" ?

*Yes, thank you! "seasonal" was changed to "altitude".*

**L601-609 and L639-641**: In these lines, the day-to-day variations of tides associated with SSWs in previous studies are being discussed. I would suggest moving this content to L653 where I see it is more relevant (with Figs. 17-19), as the current paragraphs are explaining the monthly mean amplitudes of tides.)

*The discussion in original lines 601-609 has been moved to later in this section where day-to-day tidal variability is discussed. We have retained the discussion on original lines 639-641 regarding DE3 as we feel it provides necessary motivation to compare DE3.*

**L611-612**: "latitude and ... January 2010" : repetitive (L599-600)

*This has been reworded to avoid repetition.*

**L620-622**: The peak amplitude of the tide seems to appear slightly (5-10 km) below the top of each model. Would it be related to some damping mechanisms near the model top ?

*This is potentially an issue for the lower top MERRA2 and NAVGEM-HA systems, but not for WACCMX+DART and JAGUAR-DAS whose model tops are near 500 km and 150 km, respectively. Theoretically, the upward propagating component of DW1 should peak around ~80-90 km, and there is another peak in DW1 generated in-situ in the thermosphere, which we see from 110-120 km in WACCMX+DART. The top of the WACCMX+DART system is well above these altitudes, so we do not expect damping at the model top to affect this. Likewise, the top of the JAGUAR-DAS system is well above the altitudes where DW1 peaks in theory, so effects of damping near model top are negligible.*

*We have added the following text to respond to this remark, "The range of altitudes for maximum DW1 amplitudes seen in these four analyses agrees with SABER observations (Zhang et al., 2006). For MERRA-2 and possibly NAVGEM-HA, the analysis system upper boundaries are low enough that artificially damping of DW1 may occur. In JAGUAR-DAS, DW1 is dissipated above ~100 km due to the model diffusion exponentially increasing with height to mimic molecular diffusion. The differences in DW1 structure at/above 100 km between JAGUAR-DAS and WACCMX+DART are likely due to the large differences in background zonal mean zonal wind (Fig. 3)."*

**L644**: But at quite different latitudes : 35 deg S in MERRA2 and 5 deg S in JAGUAR-DAS

*We thank the referee for noting this difference. We have revised the text to indicate the different latitudes where DE3 peaks near 50 km in MERRA2 (~35ºS) and JAGUAR-DAS (~5ºS).*

**Figs. 17-19**: 1) I would suggest exchanging (b) and (c) in order to be in the consistent order with the previous figures. 2) If the vertical lines indicate the boundaries of the months, please provide this information in the caption. If they are not, please include "(Day XX)" (the x-axis value) after "early February" in L671 and L676.

*Due to the addition of a figure at the suggestion of referee 2, these are now Figs. 18-20. We have reordered the figures, described in the caption that the black vertical lines denote the boundaries of the months, and we added vertical red lines to indicate the SSW warming period from 27 Jan – 9 Feb in response to Referee 2's request on this and other figures showing time variations. We have also added confidence estimates as dotted and solid black contours in response to the comment about L679 (see below).*

**L656-659**: Could it be removed but with just referring Section 2 ?

*Yes, we have largely re-written this paragraph.*

**L660-661**: This might be removable. : repetitive (L654-655)

*Deleted.*

**L679**: Would it be possible to check the statistical confidence in the wavelet spectrum ? (like in Torrence and Compo, 1998, BAMS)

*We have added 90% and 95% confidence estimates to new Figs. 18-20 (pasted below) based on the method outlined in Torrance and Compo (1998), using an AR1 background spectrum as described in Sassi (2012). A description of the method for obtaining these confidence estimates has been added to Section 2. Confidence estimates are generated separately for each latitude and frequency, using a background spectrum whose variance corresponds to the input temperature time series at each latitude. These results are plotted on top of the temperature amplitudes in Figs 18-20 to indicate when and where the results exceed the 90% (dashed black ) and 95% (solid black contour) limits. We find that the majority of DW1 and SW2 results fall within the 95% confidence estimates. Some of the DE3 results fall below the 90% confidence estimate in the JAGUAR-DAS results, likely a result of the much coarser time resolution (6 hourly output) compared to WACCMX_DART and NAVGEM-HA.*

[Figure]

Figure 18. Latitude-time sections of the amplitude in migrating diurnal wave 1 (DW1) temperature variations at 90 km altitude from (a) NAVGEM-HA, (b) WACCMX+DART and (c) JAGUAR-DAS for Jan.-Feb.-Mar. 2010. Thin, black, solid contours are drawn every 2 K. Bold dashed and solid black contours indicate regions where results from wavelet analysis exceed 90% and 95% confidence levels. Red vertical lines in each panel denote 27 January and 9 February, as described in the text. Black vertical lines in each panel denote 1 February and 1 March.

[Figure]

Figure 19. As in Figure 18 but for the migrating semi-diurnal wave 2 (SW2).

[Figure]

Figure 20. As in Figure 18 but for the non-migrating diurnal wave 3 (DE3).

**L684**: "to convective sources" : A reference would be helpful.

*We added a citation to Forbes et al (2008) that describes DE3s relationship to convective sources.*

**L694-695**: Please rephrase this sentence.

*We have rewritten this sentence as "For diurnal wave 1 (Fig. 21a), all three analyses show largest amplitudes near the Equator; NAVGEM-HA and JAGUAR-DAS peak values are both ~7K, while the WACCMX+DART peak value is ~4K.*

**L717**: I would suggest adding "at the equator" after "50 km altitude".

*Done.*

**Throughout the text**: there exist expressions like "from 10-20 km" or "between 10-20 km". I would suggest revising these to either "from 10 to 20 km" ("between 10 and 20 km") or "at 10-20 km". The lines where I found those are : L35, 44, 162-164, 234, 238, 239, 294, 335, 365, 441, 547, 586, 589, 694, 698, 715

*We have revised the text throughout the paper as indicated above by the referee.*

**Response to Referee 2**

We thank the referee for the constructive comments. In response to the major comments, we have added a new figure depicting correlations in temperature leading up to the major SSW (now figure 9), and we have developed a new figure 20 (now figure 21), as suggested. In response to comments from both referees, we have also added a table summarizing the main characteristics of each high-altitude meteorological analysis system, including details of the gravity wave drag parameterizations used in each model. Detailed responses for all referee comments are listed below in blue italics.

**Major comment: Interhemispheric Coupling**

You show temperature and wind fields in Fig. 5, 6, 7 and 8 but the effect is hard to se. Instead, authors like those quoted by you, show correlation coefficient between temperature anomalies at different locations. This would allow not only to see the link but also to compare with such patterns from other models.

*To aid the reader to see temperature and wind changes leading up to the SSW, Figures 5, 6, 7, 8, 18, 19, and 20 now have vertical lines indicating the onset of sustained (>5 days) easterly winds in the mesosphere (January 27) and the onset of easterly winds in the stratosphere (February 9). These dates are consistent with those highlighted in previous works (McCormack et al., 2017; Jones et al., 2018; Butler et al., 2017). In response to the referee's suggestion, we added a new figure 9 (pasted below) showing correlations among zonal mean temperatures relative to 80°N latitude and 30 km altitude over this disturbance time period. The correlation patterns among the 4 analyses are quite similar globally.*

[Figure]

Figure 9. Latitude-altitude cross-sections of the correlation coefficient between daily zonal mean temperature at 30 km and 80°N and daily zonal mean temperature at all other latitudes and altitudes in NAVGEM-HA, MERRA-2, JAGUAR-DAS, and WACCMX+DART. The SSW disturbance time period over which the correlation coefficient is calculated is from 27 January to 9 February.

**Major comment: Major-warming anomaly**
In the latitude-time plots (Fig. 17, 18 and 19) the impact of the major warming cannot be clearly identified as you have noted. In order to quantify this, I suggest to indicate the amplitudes of the tides at the time of this event in Fig. 20. Maybe, it is more adequate to take an average over the week after the central date. Such a presentation would support the aim of the study.

*As noted above, we have added lines on original Figures 17-19 indicating the stratospheric and mesospheric disturbance time period. Following the referee's suggestion to show the amplitudes of the tides over this time, we include a new figure (#21, see below) that shows the mean values and corresponding standard deviations of DW1, SW2, and DE3 amplitudes from 27 January to 9 February in the 3 analyses that extend to 90 km altitude (NAVGEM-HA, WACCMX+DART, and JAGUAR-DAS). This figure replaces the original Figure 20 and makes the point that during the time leading up to SSW onset, WACCMX+DART has smaller DW1 and DE3 amplitudes compared to NAVGEM-HA and JAGUAR-DAS, but the SW2 amplitudes are generally in good agreement among the 3 analyses.*

[Figure]

Figure 21. Latitude dependence of the mean amplitude and ±1 standard deviation values for (a) migrating diurnal wave 1 (DW1), (b) migrating semidiurnal wave 2 (SW2), and (c) non-migrating diurnal wave 3 (DE3) at 90 km altitude obtained from NAVGEM-HA, WACCMX+DART, and JAGUAR-DAS from January 27 to February 9 from Figs. 18, 19, and 20.

**Technical Comments:** In the data-and-methods section, please include a brief remark on gravity wave parameterizations for each model system. As you later speculate on this detail as one of the reasons for differences between the models.

*We have added a table listing the gravity wave drag parameterizations used within each system (see response to referee #1).*

L159: Do you mean "zonal" wind with "horizontal" wind?

*Yes this has been corrected.*

L170: "NWP" was already defined in L78.

*We now define NWP only once.*

L221: As you later refer to "JAGUAR-DAS" I would also use here this abbreviation including "DAS".

*We have revised the title of this section to be "JAGUAR-DAS".*

L423: At this point I recommend to quote Zülicke et al. (2018) who study this correlation in observations and models.

*We have added a reference to Zülicke et al. (2018) here and in the discussion of the new correlation pattern figure (see above).*

L584: Where do I find Fig. 2A? Shouldn't it be the supplemental "S2" instead?

*Yes, thank you for catching this error! Fig. 2A changed to Fig. S2.*

---

## Author Response (AR2)

*We thank the reviewers for their attention to detail and helpful comments that have improved the quality of this manuscript.*

**Response to Referee 1**

- Fig. 9 : Could you please check the colorbar ?
In the figure, the zero contours seem to be placed between the white and the lightest blue shading. Also, the place of the r = 1 label below the colorbar seems to be inconsistent with what the figure shows: this would have been placed between the most and second most dark red colors, and it might be the reason why the most dark red shading does not appear in the figure even at 30 km, 80N where r = 1.
(By the way, it is nice to see this consistency in the time variations from this new figure !)

*Thank you for noticing this detail. The colorbar was removed since all contour lines are labeled. Solid (dashed) black contours that represent positive (negative) correlations are labeled every 0.2 and this is indicated in the caption. Black filled symbols have been added to denote where r=1 at 80N and 30km (one gridpoint); this is also indicated in the caption.*

- Fig. 18 caption : Please exchange the names in (b) and (c) in the caption too.

*Thank you for catching this error! Done.*

- Fig. 21 : The magnitudes of the standard deviations (SDs) in this new figure seem to be largely different from those shown in Fig. 20 of the previous manuscript. For example, for DW1 at the equator, the SDs have magnitudes of 1 or 2 K in the new figure, whereas they were much larger in the previous figure (up to 5 K).
(I assume the difference between the solid and dashed lines indicates the SD.)
Thus, could you please confirm whether the result in the new figure is correct ?

*In Fig 21, the values of the standard deviations for the mean amplitudes of DW1, SW2, and DE3 are indeed different from the values plotted in the original Fig 20. In the new Figure 21, the mean and standard deviations are computed for the SSW period only (21 Jan – 4 Feb 2010, marked as red vertical lines), as suggested by the reviewers. In the original Figure 20, the standard deviations were plotted for the entire Jan-Feb-March 2010 period. In the case of DW1, the standard deviations are in fact larger as a result of increasing variability in the tides approaching the equinox.*

- L749 : "at 5-15N ... of 6-8 K"

*Done.*

- L752 : "in amplitude"

*Done.*

- L753 : "larger … not found" → "larger … than" ?

*Done.*

**Response to Referee 2**

L27: "5 day" --> "5-day"

*Done.*

L548: The quotation is better placed after "Eswaraiah et al., 2017" (L423 in the submitted version) and it should also appear in the list of references.

*We are puzzled by this comment. The reference to Eswaraiah et al. (2017) appears on line 443 and does not (and should not) have quotes around it. This reference already appears in the reference list.*